# Expansion of interferon inducible gene pool via USP18 inhibition promotes cancer cell pyroptosis

Kei-ichiro Arimoto[1], Sayuri Miyauchi[1], Ty D. Troutman[2], Yue Zhang[3], Mengdan Liu[3], Samuel A. Stoner[1], Amanda G. Davis[1], Jun-Bao Fan[1], Yi-Jou Huang[3], Ming Yan[1], Christopher K. Glass[2,4] & Dong-Er Zhang [1,5] ✉

While immunotherapy has emerged as a breakthrough cancer therapy, it is only effective in some patients, indicating the need of alternative therapeutic strategies. Induction of cancer immunogenic cell death (ICD) is one promising way to elicit potent adaptive immune responses against tumor-associated antigens. Type I interferon (IFN) is well known to play important roles in different aspects of immune responses, including modulating ICD in anti-tumor action. However, how to expand IFN effect in promoting ICD responses has not been addressed. Here we show that depletion of ubiquitin specific protease 18 (USP18), a negative regulator of IFN signaling, selectively induces cancer cell ICD. Lower USP18 expression correlates with better survival across human selected cancer types and delays cancer progression in mouse models. Mechanistically, nuclear USP18 controls the enhancer landscape of cancer cells and diminishes STAT2-mediated transcription complex binding to IFN-responsive elements. Consequently, USP18 suppression not only enhances expression of canonical IFN-stimulated genes (ISGs), but also activates the expression of a set of atypical ISGs and NF-κB target genes, including genes such as Polo like kinase 2 (*PLK2*), that induce cancer pyroptosis. These findings may support the use of targeting USP18 as a potential cancer immunotherapy.

Immunotherapy has emerged as a powerful tool against cancer[1]. However, despite its great potential, many cancer patients do not respond to this form of therapy[2]. A major factor contributing to initial immunotherapy resistance is insufficient presentation of tumor antigens, hindering adequate tumor T cell infiltration. Indeed, tumor antigen levels differentiate unresponsive "cold tumors" from highly immunogenic "hot tumors"[3]. Consequently, many efforts are underway to identify different therapeutic or combination approaches that could turn 'cold' tumors into 'hot' tumors, including cancer vaccines, epigenetic drugs, cytokine therapies, and adoptive T-cell transfer therapy[4].

One promising way to potentiate anti-tumor immunity against both solid tumors and hematological malignancies is induction of immunogenic cell death (ICD), a process activated by conventional therapies such as chemotherapy and radiation[5]. In addition to directly killing tumor cells, several chemotherapeutic agents, such as doxorubicin, trigger tumor-specific immune responses that promote the elimination of residual cancer cells. Furthermore, induction of the various forms of ICD (pyroptosis, ferroptosis, and necroptosis) in combination with immune checkpoint inhibitors (ICIs) synergistically enhances antitumor activity, even in *ICI*-resistant 'cold tumors'[6],

[1]Moores UCSD Cancer Center, University of California San Diego, La Jolla, CA, USA. [2]Department of Medicine, University of California San Diego, La Jolla, CA, USA. [3]Division of Biological Sciences, University of California San Diego, La Jolla, CA, USA. [4]Department of Cellular and Molecular Medicine, University of California San Diego, La Jolla, CA, USA. [5]Department of Pathology, University of California San Diego, La Jolla, CA, USA. ✉e-mail: d7zhang@health.ucsd.edu

highlighting the promise of ICD induction in overcoming this major limitation of current immunotherapies.

Mechanistically, ICD begins with the secretion of damage-associated molecular patterns (DAMPs), such as Calreticulin (CRT)[7] or High mobility group box 1 (HMGB1)[8], from dying tumor cells. DAMPs can stimulate innate immune signaling via the cyclic GMP-AMP synthase (cGAS)-STING pathway, Toll-like receptor 9 (TLR9), or Toll-like receptor 3 (TLR3), ultimately resulting in the production of type I interferon (IFN)[9]. Type I IFN binds to the IFNα receptor (IFNAR) which is composed of the IFNAR1 and IFNAR2 subunits. IFNAR then phosphorylates and activates the receptor-associated tyrosine kinases Janus kinase 1 (JAK1) and tyrosine kinase 2 (TYK2), which subsequently phosphorylate the transcription factors signal transducer and activator of transcription 1 and 2 (STAT1 and STAT2). Phosphorylated STAT1 and STAT2 hetero-dimerize and recruit IFN-regulatory factor 9 (IRF9) to form the STAT1-STAT2-IRF9 complex, which is also referred to as interferon-stimulated gene factor 3 (ISGF3). The ISGF3 complex translocates into the nucleus, where STAT2 recruits additional co-factors such as p300/CBP and binds to IFN-stimulated response elements (ISRE) to activate transcription of IFN-stimulated genes (ISGs)[10,11].

In most cases, IFNs are necessary for the subsequent immune response, but not required for the ICD itself. However, recent evidence revealed that the type I IFN response is an essential pathway for eliciting effective antitumor responses upon ICD induction[12]. For example, ICD induction by anthracycline-based chemotherapies and radiotherapy are strictly dependent on their ability to activate IFN-dependent gene expression programs in tumor cells that promote the generation of effective anti-tumor immune responses[12,13]. It has also been reported that type I IFN signaling is important for TNF, LPS, or STING-induced Necroptosis[14–16]. Regarding whether modulating the IFN response promotes ICD, it was recently shown that IFN inducible Z-form nucleic acid binding protein 1 (ZBP1) can trigger ICD in Ripk1−/− cells[17]. However, ZBP1 induction by IFN alone is insufficient to trigger ICD, suggesting that other unknown factors are required. Taken together, though many studies support the importance of the type I IFN response in ICD induction and its intrinsic ability to enhance anti-tumor immunity, it is not well characterized whether and how modulating IFN responses directly promotes ICD. Therefore, we hypothesized that there are genetic factors regulating IFN-mediated ICD. Targeting these factors may have valuable therapeutic impact considering the higher levels of interferon production in the tumor microenvironment.

Previously, we demonstrated that USP18 functions as a major negative regulator of type I IFN signaling through its binding to the IFNα receptor and STAT2[18,19]. In this study, we describe a role of nuclear USP18 in regulating cancer cell pyroptosis, a type of ICD. To define the role of USP18 in cancer, we created a Usp18 conditional knockout mouse and find that Usp18 deletion severely impairs cancer progression and improves survival. Importantly, we find that this anti-tumor effect can be exerted by the heterozygous depletion of Usp18, which does not affect normal cells. By single-cell RNAseq analysis, we also uncover evidence that inhibition of USP18 specifically targets cancer stem cells, essential for relapse. Mechanistically, we identify a nuclear role for USP18, independent of its role at the interferon receptor. Our genome wide analyses reveal that nuclear USP18 diminishes binding of IFN-regulated transcription factors to their corresponding DNA motifs in cooperation with NF-κB. This function allows USP18 to regulate both typical ISGs and non-canonical ISGs that are important for the induction of cancer cell pyroptosis. These data reveal a role for USP18 in malignancies via control of the ISG landscape and demonstrate that targeting USP18 can be an effective tool for treating cancer.

## Results

### Depletion of one *Usp18* allele does not affect normal cells but delays leukemogenesis

Among negative regulators of IFN signaling, we became interested in USP18 because low expression is correlated with better survival in several cancer types (Supplementary Fig. 1a). Furthermore, analysis of The Cancer Genome Atlas (TCGA) revealed that in fifteen out of twenty-four tumor types, USP18 expression was markedly upregulated compared to the corresponding noncancerous tissues, which may also indicate higher IFN production in several types of tumor microenvironments (Supplementary Fig. 1b). Considering these trends, we wondered whether *Usp18* depletion may enhance ICD and delay cancer development.

Prior to investigating the potential therapeutic role of *Usp18* depletion in cancer, we first determined whether this gene is a reasonable therapeutic target by testing if *Usp18* depletion affects healthy cells. In our previous study, deletion of *Usp18* in C57BL/6 mice was lethal at the embryonic stage[20]. Further analysis and comparison to Usp18+/− (He) mice revealed that *Usp18*−/− (KO) mice die around E14.5-E15.0 (Supplementary Table 1). Interestingly, *Ifnar1*−/− *Usp18*−/− mice are all alive and apparently healthy, suggesting that hyperactive type I IFN responses during embryonic development cause the lethality. At E14.5, we noticed that *Usp18*−/− mice had much smaller fetal livers than *Usp18*+/+ (WT) or *Usp18*+/− mice (Fig. 1a top). To examine this phenotype more carefully, we generated *Usp18* conditional knockout mice (*Usp18*f/f) (Supplementary Fig. 2a–c). Survival analyses of *Usp18*−/f crossed with *Usp18*+/f CMV-Cre (all tissue), UBC^ER-Cre (tamoxifen inducible expression in all tissues), and Vav-iCre (hematopoietic cell specific) revealed that the embryonic lethality of *Usp18*-deficient mice is due to defects in hematopoiesis (Supplementary Table 2 and Fig. 1a bottom).

Further analysis revealed that hematopoietic cell specific *Usp18* knockout during embryonic development and in adult mice significantly reduced hematopoietic stem cell (HSC) numbers and impaired HSC function (Supplementary Fig. 2d–p and Supplementary Table 3). Importantly, in both the embryonic and adult developmental contexts, *Usp18*+/− mice were phenotypically comparable to *Usp18*+/+ mice. We only observed an enhanced IFN response in bone marrow cells of *Usp18*+/− mice relative to *Usp18*+/+ mice (Fig. 1b), supporting the feasibility of targeting *Usp18* as a potential cancer therapy.

Having demonstrated that heterozygous *Usp18* loss does not harm healthy hematopoietic cells, we next tested the effect of heterozygous *Usp18* depletion in two well established murine leukemia models since ICD induction by chemotherapy has been shown to effectively kill leukemia cells[21]. We transduced hematopoietic cells from *Usp18*+/f UBC^ER-Cre mice with retroviruses expressing AML1-ETO9a (AE9a)-GFP or MLL-AF9 (MF9)-GFP, transplanted these cells into recipient mice, and activated Cre (Supplementary Fig. 3a). We first evaluated whether heterozygous *Usp18* deletion sensitized myeloid leukemia cells to IFN stimulation. Indeed, *Usp18*+/Δ AE9a leukemia cells were more sensitive to low amounts of type I IFN stimulation than primary bone marrow-derived macrophages (BMDM) in vitro (Fig. 1c). Next, we checked whether *Usp18* depletion could reasonably promote ICD. We observed a higher percentage of Annexin V+ cells in GFP+Lin−c-Kit+*Usp18*+/Δ leukemia cells (Fig. 1d and Supplementary Fig. 12a). Furthermore, higher numbers of CD8+ T cells and activated CD8+ T cells in viable GFP− host splenocytes in recipients of *Usp18*+/Δ cancer cells were detected. We did not observe significant enhancement of Tregs and B cells (Fig. 1e and Supplementary Fig. 12b). Both parameters support a possible increase in ICD-mediated immune responses.

We next assessed whether *Usp18* depletion can delay cancer development. Recipients of *Usp18*+/Δ AE9a or MF9 cells showed significantly delayed leukemogenesis compared to correlated controls (Supplementary Fig. 3b, c). This survival benefit was due to *Usp18*

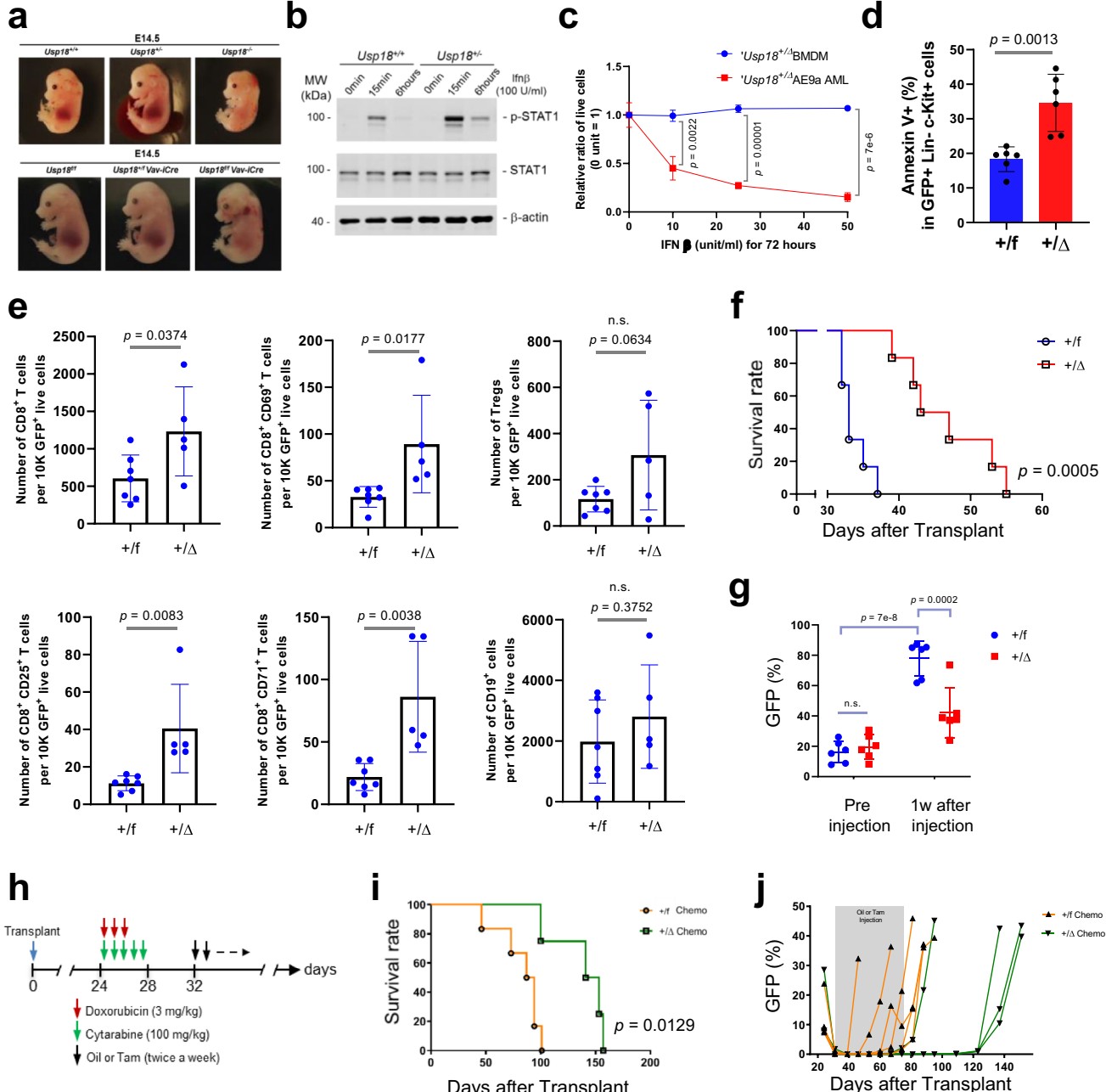

**Fig. 1 | Reduction of *Usp18* delays cancer progression. a** Representative pictures of embryos (E14.5) of *Usp18*[+/+], *Usp18*[+/−], and *Usp18*[−/−] (*Top*) and *Usp18*[fl/fl], *Usp18*[+/fl] *Vav-iCre*, and *Usp18*[fl/fl] *Vav-iCre* (*Bottom*) mice. **b** Western blots showing phosphorylated and total STAT1 protein and β-actin in *Usp18* WT and Heterozygous bone marrows. **c** Ratio of live cells from *Usp18*[+/Δ] BMDM or *Usp18*[+/Δ] AE9a cells treated with different doses of IFNβ for 72 h in one representative set of three independent repeats. *p*-value was determined by one-way ANOVA test. (*Usp18*[+/Δ] BMDM n = 3, *Usp18*[+/Δ] AE9a n = 3). **d** Percentage of Annexin V[+] GFP[+]Lin[−]c-Kit[+] splenocytes from *Usp18*[+/fl] and *Usp18*[+/Δ] AE9a recipient mice as analyzed by FACS. *p*-value was determined by two-tailed Student *t*-test. (*Usp18*[+/fl] n = 6 mice, *Usp18*[+/Δ] n = 6 mice). **e** Absolute numbers of CD8[+], activated T cells (CD8[+]/CD69[+], CD8[+]/CD25[+], CD8[+]/CD71[+]), Tregs (CD4[+]CD25[+]Foxp3[+]) and B cells (CD19[+]) in viable host splenocytes per 10,000 GFP[+] cells from *Usp18*[+/fl] and *Usp18*[+/Δ] AE9a recipient mice. Information on other immune cell populations is also shown in source data. *P*-value was determined by two-tailed Student *t*-test. (*Usp18*[+/fl] n = 7 mice, *Usp18*[+/Δ] n = 5 mice). **f** Kaplan−Meier survival curve for recipients of *Usp18*[+/fl or +/Δ] *UBC*[ER]*-Cre* AE9a-GFP AMLs. *p*-value was determined by log-rank test. (*Usp18*[+/fl] n = 6 mice, *Usp18*[+/Δ] n = 6 mice). **g** Percentage of GFP[+] cells in the PB of mice from (**f**). *p*-value was determined by one-way ANOVA test. **h** Timeline of treatments combining chemotherapy with heterozygous deletion of *Usp18* to treat AML. **i** Kaplan−Meier survival curve for mice treated as in (**h**). (Oil (+/f) n = 6 mice, Tam (+/Δ) n = 4 mice). *p*-value was determined by log-rank test. **j** Percentage of GFP[+] cells in the PB of mice from (**i**), analyzed weekly by FACS. All data represent mean ± s.d, except where indicated. n.s. = not statistically significant. Source data are provided as a Source Data file.

depletion, not Cre expression or Tamoxifen injection as neither of these factors alone affected survival in the AE9a mice model (Supplementary Fig. 3d, e). We next examined whether targeting *Usp18* can suppress established leukemia, which better represents how a therapy would be applied in a clinical setting. *Usp18*[+/fl] leukemia cells were transplanted into recipients and Cre was activated after the mice became sick. Again, *Usp18* depletion had a significant survival benefit and fewer cancer (GFP[+]) cells were observed in the peripheral blood (PB) of *Usp18*[+/Δ] mice compared to their respective controls (Fig. 1f, g). Since USP18 is a negative regulator of interferon signaling[18], we tested whether the anti-cancer effect of *Usp18* depletion is dependent on the IFN response. Using an IFNα receptor 1 (IFNAR1) blocking antibody or

IgG1 control, we compared leukemia development in *Usp18*[+/f] and *Usp18*[+/Δ] AE9a recipients. The survival benefit of *Usp18* heterozygous deletion was ablated when IFN signaling was disrupted (Supplementary Fig. 3f, g), confirming the link between USP18, IFN signaling, and anti-cancer phenotypes. We next assessed whether *Usp18* depletion can exert an enhanced anti-tumor effect in combination with a treatment that increases IFN production. STING agonists are a rapidly growing class of immunotherapeutic agents that promote type I IFN production. Therefore, we examined whether suppression of *Usp18* has a beneficial effect in combination with STING agonist-based immunotherapy. Interestingly, while the STING agonist DMXAA alone had no effect on overall survival, the survival benefit of *Usp18* depletion was significantly enhanced with the addition of DMXAA (Supplementary Fig. 3h, i), supporting the therapeutic benefit of this combination therapy. Finally, we examined whether suppression of *Usp18* has a beneficial effect in combination with chemotherapy. After confirming remission, mice transplanted with *Usp18*[+/f]*UBC*[ER]-*Cre* AE9a were injected with tamoxifen (Fig. 1h). The survival of chemotherapy-treated recipients with *Usp18*[+/Δ] leukemia cells was significantly prolonged compared to recipients of *Usp18*[+/f] cells (Fig. 1i). GFP[+] leukemia cells in the PB of *Usp18*[+/f] recipients came back earlier than those in *Usp18*[+/Δ] recipients (Fig. 1j), indicating that *Usp18* depletion delays relapse after chemotherapy. Interestingly, single-cell RNA sequencing (scRNA-seq) analysis of AML patient cells[22] revealed that expression of *USP18* was substantially elevated soon after chemotherapy induction, which is not dependent on global ISG induction (Supplementary Fig. 3j). The expression of *Usp18* was also increased after chemotherapy in our AE9a AML mouse model (Supplementary Fig. 3k). These observations suggest that residual leukemia cells may upregulate *USP18* expression as a protective mechanism against chemotherapy-induced death.

Taken together, these results indicate that reduction of *Usp18* expression potently suppresses cancer development and induces an anti-tumor immune response in vivo by a mechanism dependent on type I IFN signaling. Importantly, partial *Usp18* loss does not disrupt normal hematopoiesis, confirming a possible therapeutic window for targeting USP18.

### Heterozygous *Usp18* depletion results in loss of niche resident LSCs

To explore phenotypic variation upon *Usp18* suppression in regulating leukemogenesis, we performed scRNA-seq using GFP[+]Lin[−]c-Kit[+] *Usp18*[+/f] and *Usp18*[+/Δ] leukemia cells from recipients (Fig. 2a). First, we generated a UMAP (*Uniform Manifold Approximation and Projection*) (Fig. 2b) and confirmed higher *Isg15* expression in all clusters upon *Usp18* suppression, indicative of enhanced IFN signaling (Supplementary Fig. 4a). Next, we noticed that three populations (Clusters 8, 9, 11) were significantly altered in transplanted cancer cells after *Usp18* suppression (Fig. 2b and Supplementary Fig. 4b). Clusters 8 and 11 were expanded upon *Usp18* suppression, characterized by enhanced inflammatory signaling and ISG activation, respectively. Among inflammatory signaling pathways, IL-33 is one of the IL-1 family cytokines that is a well-known "alarmin," enhancing immunogenicity[23]. By contrast, cluster 9, characterized by the pathway including enhanced inflammasomes and/or pyroptosis, a type of ICD, signaling, was significantly depleted (Fig. 2b, c and Supplementary Fig. 4b–d). Reactome analysis revealed that differentially expressed genes (DEG) in cluster 9 were highly correlated with the DNA-damage response, NF-κB activation, and apoptosis pathways. Notably, upregulated genes were also correlated with pyroptosis (AIM2 inflammasome), a type of ICD (Fig. 2d). Interestingly, average gene expression in all clusters also show activation of inflammasome/pyroptotic related genes (Fig. 2, GSE165425) upon *Usp18* depletion. These data suggest that depleting *Usp18* may induce phenotypic features indicative of ICD.

We also noticed that cluster 9 contained the leukemia stem cell (LSC) population as feature genes were highly correlated with reported LSC feature genes[24,25] (Fig. 2e). We further conducted 'cellHarmony' analysis to identify the types of cells in our scRNA-seq dataset. Notably, immature cell types (HSCP/MPP, and Multi-lin) were most abundant in cluster 9 and substantially decreased by *Usp18* depletion (Fig. 2f). Consistent with this reduction in immature cell types, we observed reduced frequencies of both *Cd34* and *Kit* expressing cells in cluster 9 after *Usp18* depletion (Supplementary Fig. 4e). One critical component for LSCs is adhesion to the microenvironment[26,27], mediated by the interaction of Very-Late-Antigen 4 (VLA-4; composed with *Itga4* and *Itgb1*) and VCAM-1[28]. Interrogation of the feature genes in our scRNA-seq clusters revealed that cluster 9 had the highest expression of VLA-4 (Fig. 2g), further confirming LSC presence in this cluster. Furthermore, we found these VLA-4 expressing niche resident cells in cluster 9 were substantially depleted upon *Usp18* suppression in vivo (Fig. 2h). Finally, scRNA-seq analysis of GFP[+]Lin[−]c-Kit[+]*Usp18*[f/f] vs [Δ/Δ] leukemia cells showed similar cluster alterations, ensuring the reproducibility of our conclusions (Supplementary Fig. 4f–j).

Taken together, these data suggest that reduced *Usp18* expression results in loss of leukemia cells including cells with features of niche resident LSCs, which may be via apoptosis/pyroptosis, activating immune/inflammatory responses. Since LSCs are difficult to eradicate by standard chemotherapy, our phenotypic finding is important.

### Depletion of USP18 induces expansion of ISGs

Because the scRNAseq data showed interesting phenotypic variations during *Usp18* depletion and USP18 is a negative regulator of IFN signaling, we next examined how targeting USP18 alters the global ISG landscape. We performed bulk RNA-seq analysis of GFP[+]Lin[−]cKit[+] *Usp18*[+/f] and *Usp18*[+/Δ] leukemia cells. Notably, several upregulated genes have never been reported as ISGs, including inflammatory-related genes such as *Il10 and S100a8* (Fig. 3a). Gene set enrichment analysis (GSEA) revealed enrichment of pathways associated with IFN, irradiation response, TNF-signaling, and apoptosis. These enriched pathways suggest that suppression of *Usp18* induces not only IFN signaling genes but also NF-κB responsive genes (Fig. 3b). Importantly, pathway analyses suggested enhanced pyroptosis/inflammasome activation in Lin[−]cKit[+] *Usp18*[+/Δ] leukemia cells (Fig. 3c), supporting our scRNA-seq data. We further validated upregulation of IFN and irradiation responses in GFP[+]Lin[−]cKit[+] leukemia *Usp18*[+/Δ] cells by qRT-PCR and western blots (Fig. 3d, e).

Considering the intriguing observation in primary cells that *Usp18* loss induced expression of many genes that have never been classified as ISGs, we next turned to cell lines to study this unique feature of *USP18* loss more clearly. We generated human leukemia THP-1 and breast cancer MDA-MB-231 *USP18*[+/−] (He) and *USP18*[−/−] (KO) cells and observed higher IFN signaling, DNA-damage response, and annexin V[+] cells upon IFN treatment (Fig. 4a–c). To examine transcriptional changes, we performed RNA-seq using parental and *USP18*[−/−] cells. Expression analysis revealed substantial induction of both typical ISGs and non-ISGs in IFN treated *USP18*[−/−] cells, resulting in a drastically altered ISG landscape (Fig. 4d, e). Expectedly, Ingenuity Pathway Analysis (IPA) revealed that 88 genes were related to typical type I IFN signaling (Heatmap cluster A; genes upregulated in both IFN treated WT and *USP18*[−/−] THP-1 cells that were not enhanced in *USP18*[−/−] 0 h vs WT 0 h). Strikingly, 412 genes were related to inflammatory signaling such as TNF, NF-κB, and IL-33 signaling, not IFN signaling (Fig. 4e), suggesting that novel ISGs by *USP18* depletion are not regulated solely by ISRE mediated transcription (Heatmap cluster D; genes upregulated in *USP18*[−/−] + IFN vs WT + IFN and *USP18*[−/−] + IFN vs *USP18*[−/−] 0 h, but not enhanced in WT + IFN vs WT 0 h and *USP18*[−/−] 0 h vs WT 0 h). Finally,

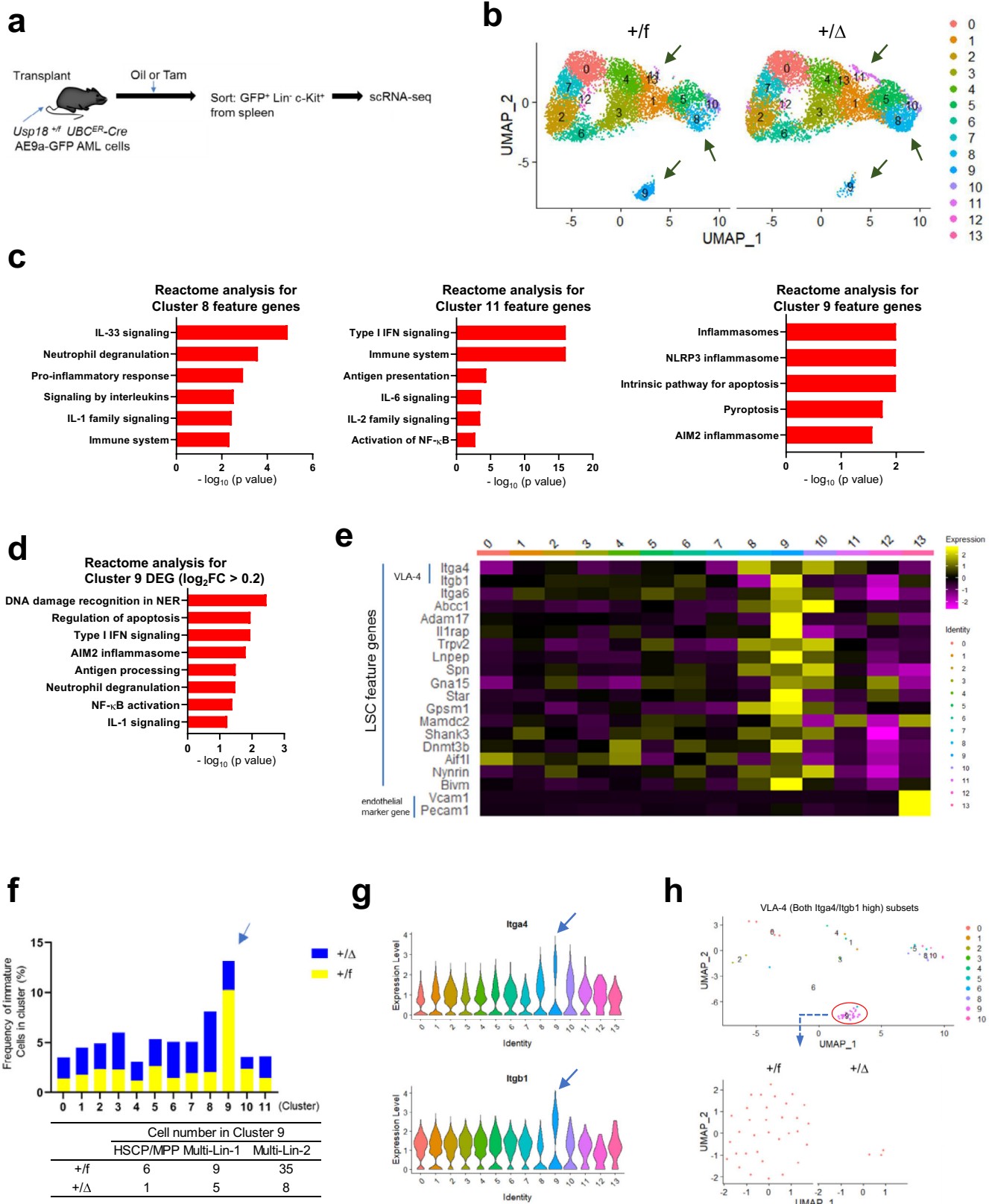

we defined genes upregulated in *USP18⁻/⁻* + IFN vs *USP18⁻/⁻* and not in WT + IFN vs WT as 'Atypical ISGs' (605 genes) and genes upregulated in both as 'Typical ISGs' (448 genes). Within atypical ISGs, 343 genes were not reported as ISGs by Interferome analysis. These genes were further divided into 'hidden atypical ISGs' or 'Non-canonical atypical ISGs' depending on the respective presence or absence of an ISRE and/or STAT1 binding motif/data in their promoter/enhancers (Fig. 4f). Again, Gene Ontology (GO) analysis showed atypical ISGs are related to the inflammatory response, not type I IFN response (Fig. 4g). Importantly, atypical ISGs were induced within 1 h of IFN treatment in *USP18⁻/⁻* cells, indicating that induction is not a secondary effect (Fig. 4h).

**Fig. 2 | Reduced *Usp18* expression results in loss of niche resident LSCs. a** Single cell RNA-seq experimental design. GFP⁺Lin⁻c-Kit⁺ cells from *Usp18*⁺/f and *Usp18*⁺/Δ AE9a recipient mice (*Usp18*⁺/f *n* = 3 mice, *Usp18*⁺/Δ *n* = 3 mice) were sorted and pooled, and then scRNA-seq was performed. **b** UMAP of sorted GFP⁺Lin⁻c-Kit⁺ cells from *Usp18*⁺/f and *Usp18*⁺/Δ AE9a recipient mice. Allows indicate the clusters that significantly changed by *Usp18* depletion. Arrows indicate clusters 8, 9, and 11. **c** Reactome analyses for feature genes of clusters 8, 9, and 11. **d** Reactome analysis for differentially expressed genes (DEG) in cluster 9. **e** Expression heat map for LSC feature and endothelial marker genes across all clusters from (**b**). **f** Frequencies of immature cells (HSCP/MPP, Multi-lin1, and Multi-lin2) in clusters with cell numbers >10, as defined by cellHarmony cell profiling analysis. Arrow indicates cluster 9. **g** Violin plots for *Itga4* and *Itgb1* in all clusters. Arrow indicate cluster 9. **h** Sub-clustering of VLA-4 expressing AML cells. VLA-4 high cells were mapped (*top* panel). Cluster 9 cells were then sub-clustered by *Usp18*⁺/f or *Usp18*⁺/Δ (*bottom* panel). *p*-values for reactome analysis in (**c**, **d**) were determined by two-sided binomial test adjusted with Benjamini−Hochberg approach. Source data are provided as a Source Data file.

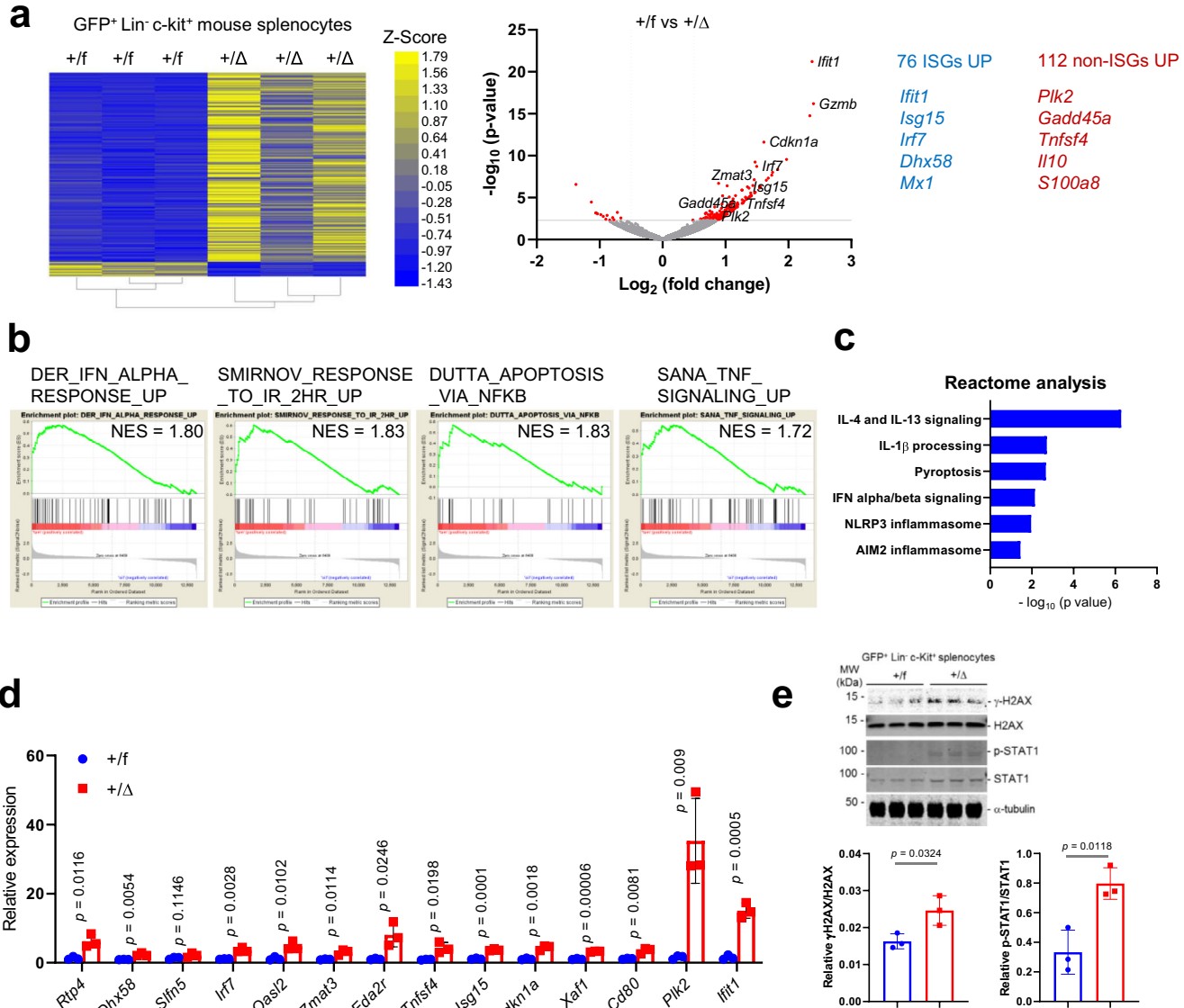

**Fig. 3 | Reduced *Usp18* expression in murine leukemia cells alters the ISG landscape. a** Heat map and volcano plot of bulk RNA-seq data from GFP⁺Lin⁻c-Kit⁺ splenocytes from *Usp18*⁺/f and *Usp18*⁺/Δ AE9a recipients. (*Usp18*⁺/f *n* = 3 mice, *Usp18*⁺/Δ *n* = 3 mice) (log₂ fold change >1 (Heatmap) or >0.5 (Volcano) and *p*-value <0.005). *p*-value was determined by two-sided Wald test adjusted with Benjamini−Hochberg approach. **b** Representative GSEA plots from RNA-seq data of (**a**). **c** Reactome analysis from RNA-seq data of (**a**). *p*-value was determined by two-sided binomial test adjusted with Benjamini-Hochberg approach. d; qRT-PCR analysis of IFN and DNA damage inducible genes in sorted GFP⁺Lin⁻c-Kit⁺ splenocytes from *Usp18*⁺/f and *Usp18*⁺/Δ AE9a recipient mice. (*Usp18*⁺/f *n* = 3 mice, *Usp18*⁺/Δ n = 3 mice). Data represent mean ± s.d. **e** Western blot analysis of sorted GFP⁺Lin⁻c-Kit⁺ splenocytes from *Usp18*⁺/f and *Usp18*⁺/Δ AE9a recipient mice (*Usp18*⁺/f *n* = 3 mice, *Usp18*⁺/Δ *n* = 3 mice). The bar graphs show the indicated protein ratios. The protein samples ran separate gels because total and phosphorylated STAT1 or H2AX are same molecular weight. *p*-value was determined by two-tailed Student *t*-test. Data represent mean ± s.d. Source data are provided as a Source Data file.

Overall, these data demonstrate that depletion of *USP18* drastically alters the ISG transcriptional landscape in the tumor microenvironment in vivo or upon IFN treatment in cell lines. Considering this finding, we hypothesized that USP18 may mediate atypical ISG expression by regulating transcriptional machinery beyond the canonical STAT2-ISRE interaction.

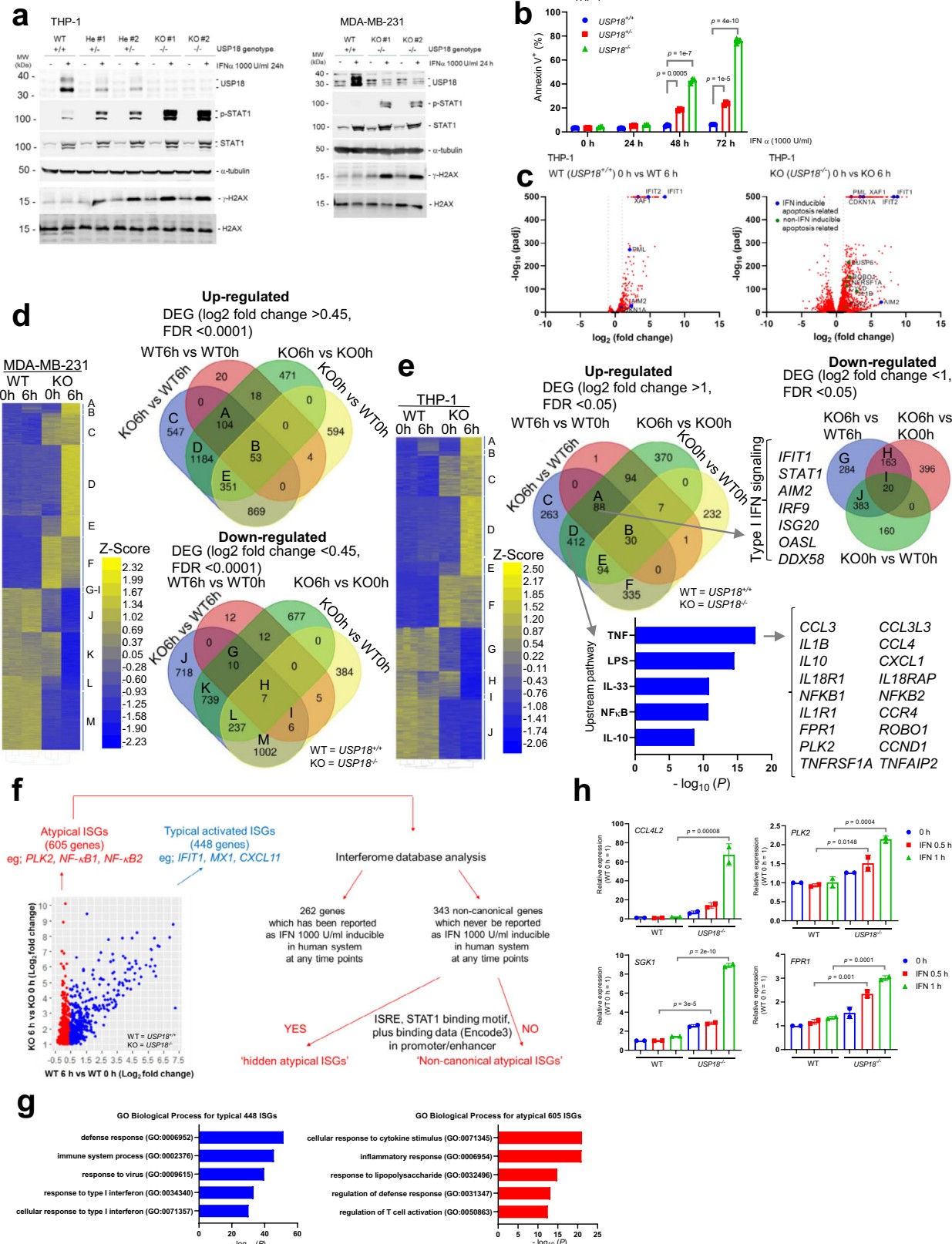

## USP18-STAT2 epigenetically controls ISG expansion

Having identified a subset of atypical ISGs that are only activated when IFN treatment is combined with *USP18* deficiency, we next sought to uncover the molecular mechanism for this. USP18 has previously been described to inhibit IFN signaling at the interferon alpha/beta receptor (IFNAR2). However, utilizing a proximity-

dependent-biotin-labeling (BioID) approach to identify proximal proteins upon expression of USP18, we uncovered numerous nuclear-localized proteins including transcription factors (TFs) and chromatin modifiers (Supplementary Fig. 5a). Therefore, we hypothesized that USP18 plays an undiscovered nuclear role, directly regulating transcriptional machinery.

**Fig. 4 | USP18 loss in human cancer cells alters the landscape of ISGs. a** Western blots depicting proteins related to IFN-signaling and the DNA damage response in WT, *USP18*[+/−] (2 different clones) and *USP18*[−/−] (2 different clones) THP-1, and WT, and *USP18*[−/−] (2 different clones) MDA-MB-231 cells. **b** Percentage of annexin V[+] THP-1 cells (WT, *USP18*[+/−] or *USP18*[−/−]) when treated with IFNα (1000 U/ml) for the indicated times. (*n* = 3 independent samples each). Data represent mean ± s.d. **c** Volcano plot of RNAseq data comparing WT vs WT + IFNα and *USP18*[−/−] (KO) vs KO + IFNα (1000U/ml) THP-1 cells (*n* = 3 independent samples each). *p*-value was determined by two-sided Wald test adjusted with Benjamini–Hochberg approach. **d** Heat map representing RNAseq data for MDA-MB-231 WT and *USP18*[−/−] cells with or without IFNα (1000 U/ml) for 6 h (*n* = 3 independent samples each). **e** Heat map representing RNAseq data from THP-1 WT and *USP18*[−/−] cells with or without IFNα treatment for 6 h. IPA pathway analysis and representative genes in clusters A and D are shown. *p*-value was determined by right-tailed Fisher's exact test adjusted with Benjamini–Hochberg approach. **f** Dot plot and schematic representing how ISGs

were classified as 'typical' versus 'atypical'. Genes upregulated in *USP18*[−/−] + IFNα vs *USP18*[−/−] and not in WT + IFNα vs WT were labeled 'Atypical ISGs' (605 genes) and genes upregulated in both were labeled 'Typical ISGs' (448 genes). Within atypical ISGs, 262 genes have been reported as IFNα (1000 U/ml) inducible in human cells. However, 343 genes are not reported as ISGs by Interferome analysis. These genes were further divided into 'hidden atypical ISGs' or 'Non-canonical atypical ISGs' depending on the respective presence or absence of an ISRE and/or STAT1 binding motif/data in their promoter/enhancers. **g** Typical ISGs and atypical ISGs were analyzed by GO biological process analysis. *p*-value was determined by one-sided Fisher's exact test adjusted with Benjamini–Hochberg approach. **h** qRT-PCR analysis of WT and *USP18*[−/−] THP-1 cells (*n* = 2 independent samples each) treated with or without IFNα (1000 U/ml) for indicated times. Data represent mean ± s.d. *p*-value was determined by one-way ANOVA test. Source data are provided as a Source Data file.

To address this, we performed a quantitative genome wide comparison of active promoter/enhancer elements in WT or *USP18*[−/−] THP-1 cells with and without IFN treatment. Specifically, we generated and then integrated ATAC-seq data with CHIP-seq data for H3K27ac. IFN treatment of parental cells had negligible impact on chromatin accessibility and minimal impact on altered activity (Fig. 5a and Supplementary Fig. 5b). Conversely, USP18 deficient cells had increased accessibility and increased H3K27ac at promoter/enhancers, effects that were both augmented by IFN treatment (Fig. 5a and Supplementary Fig. 5b). These findings indicate that USP18 is involved in large-scale regulation of the epigenetic landscape and global gene regulatory responses to IFN treatment.

To predict mechanisms associated with this regulation, we profiled ATAC-seq peaks with increased accessibility or acetylation for enrichment of TF binding motifs. We first observed more significant enrichment of the ISRE motif in IFN treated *USP18*[−/−] cells than in IFN treated WT cells (Fig. 5b and Supplementary Fig. 5c, d). This observation aligns with the expanded list of atypical ISGs in the *USP18*[−/−] context, some of which do indeed harbor an ISRE sequence ('hidden ISGs').

Since we previously found that USP18 binds to STAT2[19], we hypothesized that USP18 may disrupt STAT2/IRF9 complex formation or binding to target DNA sequences. Neither USP18 alone nor in combination with ISG15 (which stabilizes USP18)[29] affected the interaction between STAT2 and IRF9 (Supplementary Fig. 6a, b). Additionally, USP18 cannot bind IRF9 without STAT2 (Supplementary Fig. 6c). Next, DNA pull-down analysis revealed that IRF9 and STAT2 interacted only with the WT ISRE and this binding was substantially reduced upon USP18 expression (Fig. 5c left). However, IRF9-ISRE binding was unchanged by USP18 expression in STAT2−/− cells (Supplementary Fig. 6d), confirming that this mechanism is STAT2-dependent. We further verified that USP18 does not affect the binding between GAS (Gamma IFN activation site) and STAT1, nor does it impact ISRE-IRF7 binding (Supplementary Fig. 6e, f). USP18 was also able to inhibit IRF9/STAT2 binding to IRE sequences (minimum ISRE element 'TTTC') (Fig. 5c right), present in 'non-canonical' atypical ISGs that lack ISRE motifs. In reporter assays, USP18 expression significantly decreased ISRE and IRE-mediated luciferase activation by IRF9/STAT2 or ISGF3 (Fig. 5d). ChIP-qPCR analysis of several atypical ISGs showed a clear reduction of IRF9 occupancy to promoter/enhancer regions by USP18 ectopic expression in THP-1 cells (Supplementary Fig. 7a). We further validated one of these atypical ISGs, Polo like kinase 2 (PLK2), via a reporter assay. PLK2 enhancer activity was increased with IRF9/STAT2 expression but decreased with USP18 co-expression (Supplementary Fig. 7b). These findings demonstrate that USP18 reduction induces ISG diversity by permitting IRF9/STAT2 binding to ISRE and IRE motifs in promoters/enhancers of atypical ISGs.

To determine whether other TFs may contribute to this newly ascribed nuclear role of USP18 in modulating the ISG landscape, we returned to our TF motif enrichment analysis (Fig. 5b). Among

additional motifs, we became interested in NF-κB because we observed activation of this pathway in AE9a LSCs and THP-1 cells following *USP18* suppression (Figs. 2d and 4e). Indeed, NF-κB related regulators, not IFN, were top upstream regulators of atypical ISGs by Ingenuity Pathway Analysis (IPA) (Supplementary Fig. 8a). This observation was confirmed by ChIP-Atlas-enrichment-analysis showing enrichment of NF-κB (p65) binding to promoters/enhancers of atypical ISGs (Supplementary Fig. 8b). Therefore, we first verified the presence of NF-κB motifs in active promoters/enhancers of both hidden (*PLK2*) and non-canonical atypical ISGs (*CCL4L2*, *SGK1*, and *FPR1*) previously described to have anti-cancer functions[30–33] (Fig. 5e and Supplementary Fig. 8c). Next, we tested whether p65 is important for atypical ISG promoter activity. Indeed, *CCL4L2* promoter activity is enhanced by ISGF3 expression, decreased by USP18 co-expression, and substantially abrogated if the NF-κB motif is deleted (Supplementary Fig. 8d). Moreover, higher levels of p65 were observed in the nucleus only in IFN treated *USP18*[−/−] cells (Fig. 5f). Additionally, IRF9 and p65 occupancies in the promoter/enhancer region of *PLK2* and *CCL4L2* were significantly higher in IFN treated *USP18*[−/−] cells than in IFN treated WT cells (Fig. 5g). Furthermore, the NF-κB inhibitor, GYY4137, reduced the induction of *PLK2* and *CCL4L2* in IFN treated *USP18*[−/−] cells (Supplementary Fig. 8e). Finally, to test whether atypical ISG expression can be regulated by similar epigenetic machinery in *Usp18* depleted murine leukemia cells in vivo as in the human cell line context, we also performed paired ATAC and RNA-seq using *Usp18*[+/f] and *Usp18*[+/Δ] murine leukemia cells. Compared to the human cell lines, we observed similar gene expression alterations, but less expansion of ISGs (Supplementary Fig. 8f). For the atypical ISG *Plk2*, we couldn't see a detectable difference in ATAC-seq peaks, which was the same as IFN treated *USP18*[−/−] THP-1 cells. But we could detect a significant increase in H3K27ac activity in the promoter/enhancer of the *Plk2* gene, which has an IRE or ISRE and NF-κB binding sites, suggesting that USP18 could modulate typical and atypical ISGs via a similar epigenetic mechanism in both human and mouse contexts (Supplementary Fig. 8g). Altogether, these data demonstrate that USP18 decreases the occupancy of IRF9-STAT2-p65 complexes at ISRE/IRE/NF-κB DNA in the nucleus, a function distinct from its role in regulating IFN signaling at the interferon alpha/beta receptor (IFNAR2).

## ISG expansion by *USP18* depletion regulates immunogenic cancer cell death

We next examined the pathways uniquely affected by IFN treatment in a *USP18*[−/−] cellular context. Reactome analysis revealed that commonly upregulated genes by IFN in both THP-1 and MDA-MB-231 *USP18*[−/−] cells were related to cytokine signaling, caspase activation, inflammasome activation, pyroptosis, and IL-1β processing (Fig. 6a). Just like we observed in our scRNA-seq data of primary leukemia cells (Fig. 2c, d), these data demonstrate that lack of *USP18* may result in not only apoptosis but also pyroptosis, a form of ICD. Consistent with this

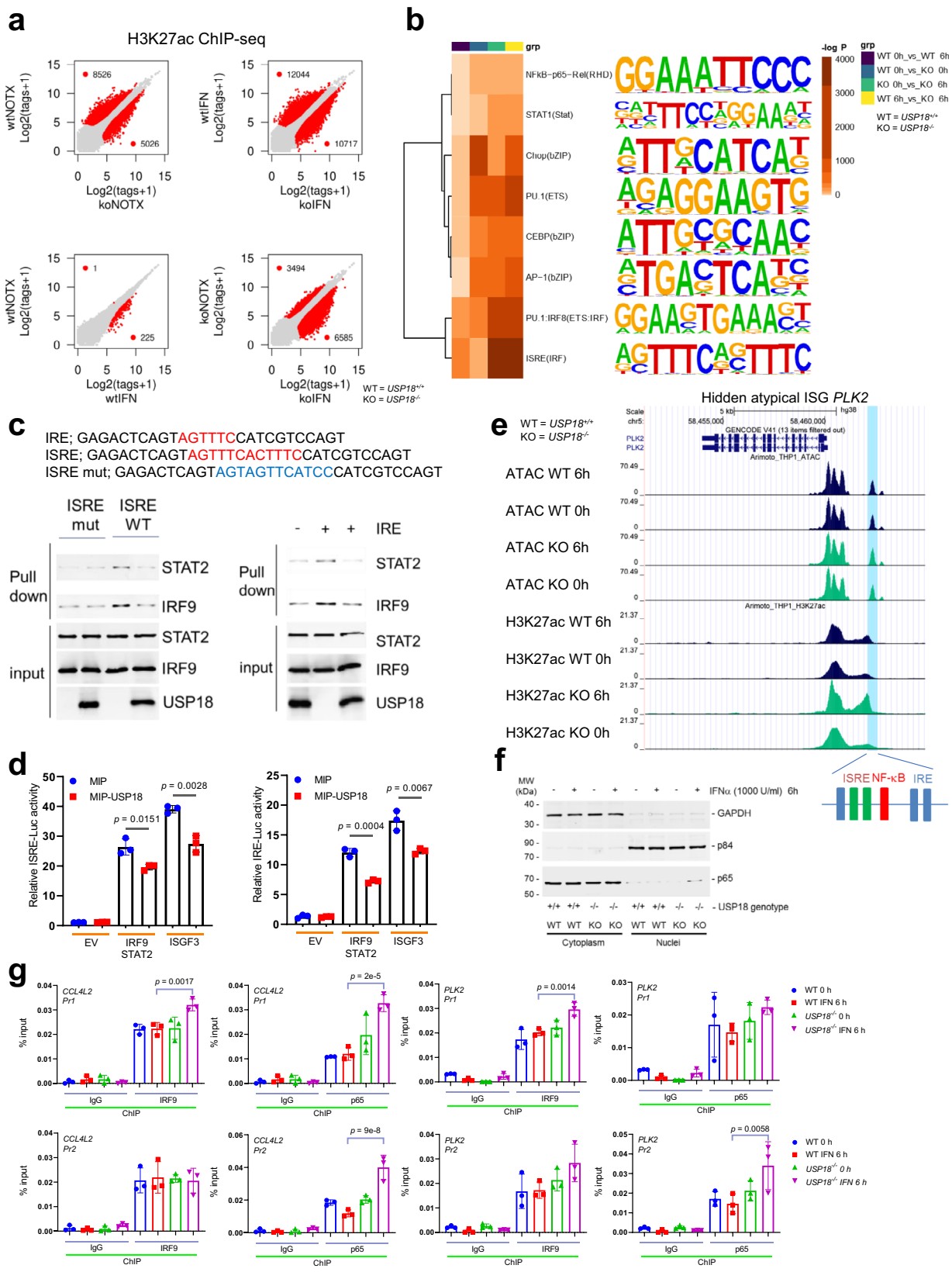

hypothesis, the secretion of IL-1β and cleaved IL-1β levels were substantially increased in the culture supernatant of IFN treated *USP18⁻/⁻* cells (Fig. 6b).

ICD is accompanied by increased membrane permeability, cell swelling, and release of cytosolic contents, such as lactate dehydrogenase (LDH). Additionally, Calreticulin (CRT), one of the DAMP

molecules, is translocated from the lumen of the endoplasmic reticulum to the surface of the dying cell (ecto-CRT)[7] upon ICD induction. Here, we observed significantly greater levels of cell death, LDH release, and ecto-CRT levels in IFN treated *USP18⁺/⁻* or *USP18⁻/⁻* THP-1 cells compared to WT (Fig. 6c, d, Supplementary Fig. 12c). We confirmed that cell death in IFN treated *USP18⁻/⁻* cells was not due to

**Fig. 5 | Nuclear USP18 regulates ISG landscape through IRF9/STAT2 and NFκB.**
**a** Scatter plots of H3K27ac tag counts at promoter/enhancer regions in WT vs.
*USP18⁻/⁻*, WT vs. WT IFN, WT IFN vs. *USP18⁻/⁻* IFN, and *USP18⁻/⁻* vs. *USP18⁻/⁻* IFN THP-1
cells (*n* = 2 independent samples each). Red data points represent peaks with >2-
fold change. **b** Heat map showing a comparative motif enrichment at enhancer
regions defined by differential H3K27ac in (**a**). *p*-value was determined by two-tailed
Hypergeometric test adjusted with Benjamini−Hochberg approach. **c** DNA pull-
down assays with the indicated DNA probes and combinations of STAT2, IRF9, and
USP18 proteins. **d** Relative firefly to renilla luciferase activity in U5A (IFNAR2⁻/⁻) cells

expressing the indicated luciferase reporter constructs and combination of pro-
teins. (*n* = 3 independent samples each). Data represent mean ± s.d. *p*-value was
determined by one-way ANOVA test. **e** Genome browser tracks depicting ATAC and
H3K27ac peaks in the atypical ISG *PLK2*. **f** Western blot analysis of the indicated
proteins in cytoplasmic and nuclear fractions of *USP18⁻/⁻* THP-1 cells with or without
IFNα (1000 U/ml) treatment. **g** IRF9 or p65 ChIP qPCR of the *PLK2* and *CCL4L2*
promoters in WT or *USP18⁻/⁻* THP-1 cells with or without IFN treatment. (*n* = 3
independent samples each). Data represent mean ± s.d. *p*-value was determined by
one-way ANOVA test. Source data are provided as a Source Data file.

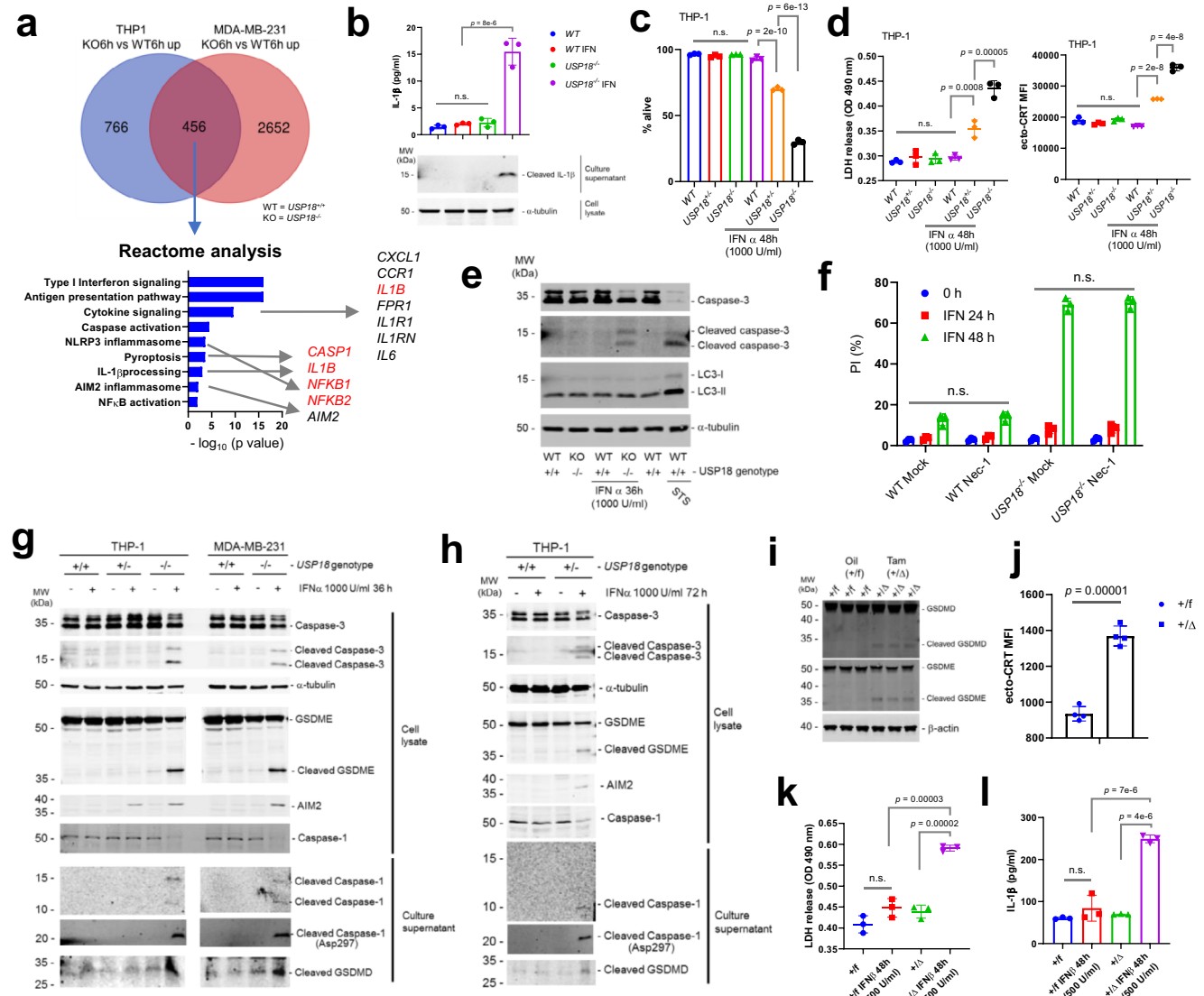

**Fig. 6 | Loss of USP18 switches IFN-induced apoptosis to pyroptosis by altering
the ISG landscape. a** Reactome analysis of commonly enhanced genes in IFNα
treated *USP18⁻/⁻* THP-1 and MDA-MB-231 cells. *p*-value was determined by two-sided
binomial test adjusted with Benjamini−Hochberg approach. **b** IL-1β secretions and
cleaved IL-1β in culture supernatant were analyzed in MDA-MB-231 WT and *USP18⁻/⁻*
cells with or without IFNα (1000 U/ml) for 36 h (*n* = 3 independent samples each).
**c** Percentage of live WT, *USP18⁺/⁻* and *USP18⁻/⁻* THP-1 cells treated with or without
IFNα (1000 U/ml) for 48 h (*n* = 3 independent samples each). **d** LDH release assay
and MFI of surface-exposed calreticulin (ecto-CRT) of WT, *USP18⁺/⁻* and *USP18⁻/⁻*
cells with or without IFNα (1000U/ml) for 48 h (*n* = 3 independent samples each).
**e** Western blot analysis of WT and *USP18⁻/⁻* THP-1 cells treated with or without IFNα
(1000U/ml) for 36 h (lane 1−4) or WT cells treated with staurosporine for 24 h (lanes
5, 6). **f** Percentage of live WT and *USP18⁻/⁻* THP-1 cells pre-treated with mock or
Necrostatin-1 (10 μM) for 6 h, then treated with IFNα (1000 U/ml) for the indicated

times (*n* = 3 independent samples each). **g** Western blot analysis of THP-1 cells (WT,
*USP18⁺/⁻* and *USP18⁻/⁻*) and MDA-MB-231 cells (WT and *USP18⁻/⁻*) treated with or
without IFNα for 36 h. **h** Western blot analysis of WT and *USP18⁺/⁻* THP-1 cells
treated with or without IFNα for 72 h. **i** Western blot of sorted GFP⁺Lin⁻c-Kit⁺
splenocytes from *Usp18⁺/f* and *Usp18⁺/Δ* AE9a recipient mice (*Usp18⁺/f* *n* = 3, *Usp18⁺/Δ*
*n* = 3 mice). **j** MFI of ecto-CRT of sorted GFP⁺Lin⁻c-Kit⁺ splenocytes from *Usp18⁺/f* and
*Usp18⁺/Δ* AE9a recipient mice (*Usp18⁺/f* *n* = 3, *Usp18⁺/Δ* *n* = 3 mice). *p*-value was
determined by two-tailed Student *t*-test. **k** LDH release assay of *Usp18⁺/f* and *Usp18⁺/Δ*
AE9a cells after 48 h of treatment with or without murine IFNβ (500U/ml) (*n* = 3
independent samples each). **l** Analysis of IL-1β secretions of culture supernatant
from (**k**). All data represent mean ± s.d, except where indicated. n.s. = not statisti-
cally significant. *p*-value for (**b**−**d**, **f**, and **k**, **l**) was determined by one-way ANOVA
test. Source data are provided as a Source Data file.

autophagy or necrosis (Fig. 6e, f). Furthermore, inflammasome-Caspase-1-mediated GSDMD cleavage and Caspase-3-mediated GSDME cleavage, unique markers of pyroptosis[34–36], were also observed in IFN treated *USP18*[+/ and –/–] cells (Fig. 6g, h). Importantly, we also detected pyroptotic phenotypes in our *Usp18*[+/ΔAE9a] mouse model, and *Usp18*[+/ΔAE9a] cell line, supporting our hypothesis in Fig. 1 (Figs. 6i–l and 12d).

ICD was originally reported in solid tumors. To test whether our findings can be translated to solid cancer models, we generated *Usp18*[+/+], *Usp18*[+/–], and *Usp18*[–/–] B16F10 melanoma and MC38 colon cancer cells, subcutaneously injected them into mice, and monitored tumor growth. We observed a clear reduction in the development of both tumor types in mice injected with *Usp18*[+/–] or *Usp18*[–/–] cells compared to those injected with *Usp18*[+/+] cells, although their growth in vitro was similar (Supplementary Fig. 9a). Furthermore, although we could not quantify the GSDMD or GSDME cleavage due to low detection of the full-length proteins in these cells, we detected significantly elevated levels of LDH release and ecto-CRT in IFN treated *Usp18*[–/–] B16F10 and MC38 (Supplementary Fig. 9b). Additionally, we found clear translocation of HMGB1 from the nucleus to the cytoplasm in *Usp18*[+/–] and *Usp18*[–/–] MC38 tumors, which is an ICD associated event seen in solid tumors (Supplementary Fig. 9c left). Indeed, higher infiltration of CD8[+] T cells was observed in these tumors, suggesting that depletion of *Usp18* in solid tumors can induce ICD and enhance the immune response (Supplementary Fig. 9c right). Finally, to further test for ICD in *Usp18*-depleted tumors, we used the gold standard vaccination assay[7,37,38] (Supplementary Fig. 10a). Mice were vaccinated with PBS only, *Usp18*[+/–] B16F10 cells treated with cisplatin (non bona-fide ICD inducer)[39] or IFN, or with or without IFN treated *Usp18*[+/+] or *Usp18*[+/–] MC38 cells 1 week prior to the challenge of respective WT cells. Strikingly, we observed a significant reduction of tumor development in the group of mice vaccinated with IFN treated *Usp18*[+/–] cells (Supplementary Fig. 10b, c). Importantly, this vaccine effect is through the enhancement of tumor-infiltrating activated CD8[+] T cells, which is critical for ICD-induced vaccine effect (Supplementary Figs. 10d, 12e). Thus, ICD induced by targeting *Usp18* occurs not only in hematological malignancies but also certain solid cancers.

## USP18 depletion induces both GSDMD and GSDME-dependent pyroptosis upon IFN treatment

Depletion of USP18 induced GSDMD and GSDME cleavage, both markers of pyroptosis, upon IFN treatment. To further characterize the pyroptosis in IFN treated *USP18*[–/–] cells, we created *GSDMD*[–/–], *GSDME*[–/–], *USP18*[–/–]*GSDMD*[–/–], and *USP18*[–/–]*GSDME*[–/–] cells (Fig. 7a). As expected, both IFN treated *USP18*[–/–]*GSDMD*[–/–] and *USP18*[–/–]*GSDME*[–/–] cells exhibited reduced ICD markers (LDH release and ecto-CRT) compared to IFN treated *USP18*[–/–] cells (Fig. 7b and Supplementary Fig. 12c). Interestingly, IFN treated *USP18*[–/–]*GSDMD*[–/–] cells showed similar levels of cell death as IFN treated *USP18*[–/–] cells; however, IFN treated *USP18*[–/–]*GSDME*[–/–] cells showed less cell death, suggesting that GSDME cleavage, not GSDMD cleavage, is significantly involved in the cell death of IFN treated *USP18*[–/–] cells (Fig. 7c).

Caspase-1 and 8 are well known to be involved in GSDMD and IL-1β cleavage. Caspase-8 can be involved in GSDME cleavage as well. To validate that the GSDMD pathway is active in IFN treated *USP18*[–/–] cells, we transduced WT and *USP18*[–/–] cells with Caspase 1 sgRNAs. Reduction of Caspase-1 in IFN treated *USP18*[–/–] cells substantially decreased the levels of GSDMD cleavage (Fig. 7d) and IL-1β secretion (Fig. 7e), indicating that the Caspase-1-GSDMD pathway is activated in IFN treated *USP18*[–/–] cells. Importantly, the reduction of Caspase-1 did not affect the cell death in IFN treated *USP18*[–/–] cells (Fig. 7f), which is consistent with the consequence of IFN treated *USP18*[–/–]*GSDMD*[–/–] cells.

Next, we examined factors upstream of the Caspase-1-GSDMD pathway in IFN-treated *USP18*[–/–] cells. We focused on NLRP3 and/or the AIM2 inflammasome because the cluster 9 cells from our scRNA-seq analysis of *Usp18*[+/Δ] primary murine leukemia cells predicted inflammasome activation (Fig. 2c). Furthermore, commonly upregulated genes by IFN in both THP-1 and MDA-MB-231 *USP18*[–/–] cells were related to NLRP3 and AIM2 inflammasome activation (Fig. 6a). Interestingly, MCC950 (NLRP3 inflammasome inhibitor) substantially inhibited the cleavage of GSDMD and IL-1β secretion in IFN treated *USP18*[–/–] cells, but ODN-A151 (AIM2 inflammasome inhibitor) did not affect either, suggesting that the NLRP3 inflammasome is activated in IFN treated *USP18*[–/–] cells (Fig. 7g, h). Again, MCC950 treatment did not affect cell death, despite the loss of GSDMD cleavage, although it remains possible that caspase-8-mediated cleavage of GSDME complements the cell death, which needs to be further investigated in future (Fig. 7i).

USP18 is an isopeptidase that cleaves the ubiquitin-like ISG15 from target proteins[40]. To test whether the enzymatic activity of USP18 affects pyroptosis signaling, we expressed empty vector, USP18 cDNA, or USP18 C64S (enzyme dead) cDNA in *USP18*[–/–] THP-1 cells. Upon IFN treatment, expression of both WT and C64S mutant USP18 similarly inhibited the cleavages of GSDMD and GSDME, cell death, LDH release, ecto-CRT, and expression of typical (*IFIT1*) and atypical (*PLK2* and *CCL4L2*) ISGs, indicating that USP18 enzymatic activity does not affect the ICD phenotypes (Fig. 7j–m and Supplementary Fig. 12c).

USP18 is also involved in many cellular processes other than IFN signaling or ISG15 deconjugation[41]. To verify that the USP18-STAT2 interaction regulates ICD, we utilized the STAT2 R148W mutant which was recently shown to decrease its binding to USP18[42]. Expressing STAT2 R148W in *STAT2*[–/–] (U6A) cells should phenocopy the USP18-depleted cells even though USP18 is intact. We first confirmed that 2fTGH cells, the parental line from which the U6A line was derived, showed elevated cell death and ICD markers upon USP18 depletion and IFN treatment (Fig. 7n–o and Supplementary Fig. 12c). Next, we expressed STAT2 R148W in the U6A cells and observed an elevated IFN response (*IFIT1* expression), as was previously reported (Fig. 7p). Upon IFN treatment of these cells, we observed increased atypical ISG *PLK2* expression, cell death, IL-1β secretion, GSDMD cleavage, LDH release, and ecto-CRT (Fig. 7p–t and Supplementary Fig. 12c).

Taken together, these results indicate that the USP18-STAT2 interaction regulates both GSDMD and GSDME-dependent pyroptosis and that the enzymatic activity of USP18 is not necessary for ICD.

## PLK2, an atypical ISG, regulates GSDME-dependent pyroptosis and leukemogenesis

Since pyroptosis was induced by *USP18* depletion but has not been well described upon interferon treatment, and Caspase-3-mediated GSDME cleavage is not involved in canonical inflammasome-Caspase-1-mediated GSDMD cleavage, we hypothesized that atypical ISGs may contribute to cleaved GSDME-mediated pyroptosis. Using our RNA-seq datasets, we identified 8 genes that were commonly upregulated in *USP18*-depleted human and murine malignant cells (Supplementary Fig. 11a). Among these genes was the atypical ISG polo like kinase 2 (PLK2), a gene that is downregulated in AML patients and thus may play a tumor suppressive role in cancer (Supplementary Fig. 11b). We confirmed that PLK2 expression is increased upon interferon treatment only in *USP18*[–/–] cells (Supplementary Fig. 11c, d). Furthermore, we also detected elevated PLK2 protein levels in *Usp18*[+/Δ] leukemia cells compared to *Usp18*[+/f] leukemia cells (Supplementary Fig. 11e). Interestingly, the MV4-11 cell line was the only cell line that did not show an increase in PLK2 and did not have elevated cell death upon IFN treatment of *USP18* depleted cells (Supplementary Fig. 11f, g). This observation supports our hypothesis that PLK2 contributes to our cell death phenotype. Moreover, PLK2 knockdown in *USP18*[–/–] cells, not WT cells, led to fewer Annexin V[+] cells upon IFN treatment (Supplementary Fig. 11h–j).

To further characterize PLK2 in *USP18*[–/–] cells, we created *PLK2*[–/–] and *USP18*[–/–]*PLK2*[–/–] cells. Importantly, the deletion of PLK2 in IFN-treated *USP18*[–/–] cells showed a clear reduction of GSDME cleavage but

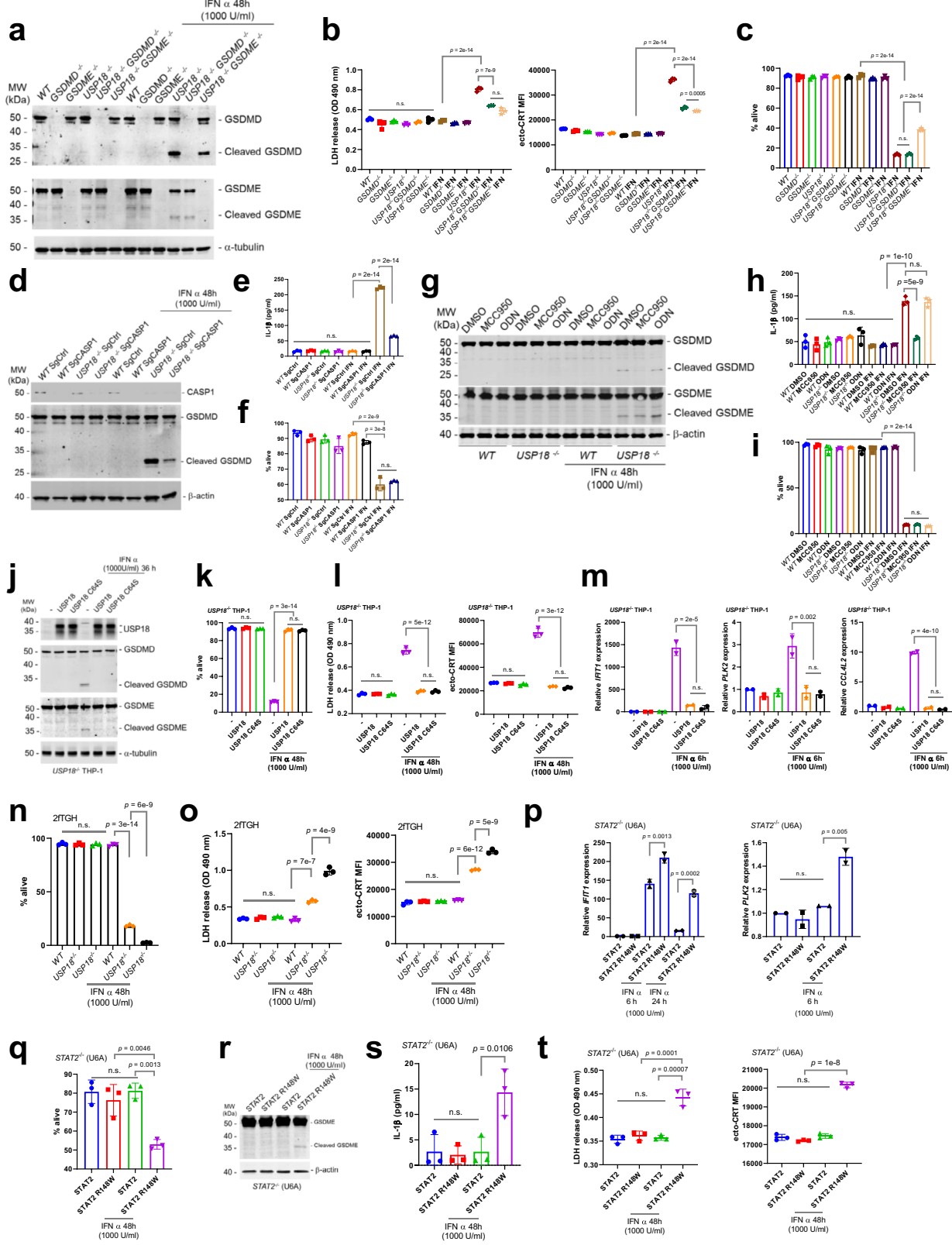

not GSDMD cleavage (Fig. 8a). PLK2 deficiency resulted in reduced cell death, LDH release, and ecto-CRT levels compared to IFN treated *USP18*$^{-/-}$ cells (Fig. 8b, c and Supplementary Fig. 12c). Strikingly, overexpression of PLK2 in PLK2$^{-/-}$ cells promoted processing of Caspase-3 and cleavage of GSDME, but did not affect cleavage of GSDMD (Fig. 8d), linking PLK2 expression to GSDME dependent pyroptosis.

Notably, PLK2 expression induced cell death, enhanced LDH release, and increased ecto-CRT but did not induce IL-1β secretion (Fig. 8e, f and Supplementary Fig. 12c), suggesting that the PLK2-GSDME pathway is totally independent of the GSDMD pathway. Furthermore, PLK2 kinase activity was not required to induce cleavage of Caspase-3 and GSDME (Fig. 8g, h). Interestingly, PLK2 enzymatic activity suppresses

**Fig. 7 | USP18-STAT2 regulates GSDMD and GSDME-dependent pyroptosis upon IFN stimulation. a** Western blot of WT, *GSDMD⁻/⁻*, *GSDME⁻/⁻*, *USP18⁻/⁻*, *USP18⁻/⁻GSDMD⁻/⁻* and *USP18⁻/⁻GSDME⁻/⁻* THP-1 cells treated with or without IFNα (1000 U/ml) for 48 h. **b** Analysis of ICD markers in (**a**). **c** Percentage of live cells in (**a**) analyzed by FACS. **d** Western blot of WT or USP18⁻/⁻ cells expressing Control sgRNA or Caspase-1 sgRNA after treatment with IFNα (1000 U/ml) for 48 h. **e** Analysis of IL-1β secretions of culture supernatant in (**d**). **f** Percentage of live cells in (**d**) analyzed by FACS. **g** Western blot of DMSO, MCC950 or ODN-A151 treated WT or *USP18⁻/⁻* THP-1 cells further treated with or without IFNα (1000 U/ml) for 48 h. **h** Analysis of IL-1β secretions of culture supernatant in (**g**). **i** Percentage of live cells in (**g**) analyzed by FACS. **j** Western blot of *USP18⁻/⁻* THP-1 cells expressing Mock, USP18 or USP18C64S treated with or without IFNα for 48 h. **k** Percentage of live cells in (**j**) analyzed by FACS. **l** Analysis of ICD markers in (**j**). **m** qRT-PCR of *IFIT1*, *PLK2* and *CCL4L2* expression in *USP18⁻/⁻* THP-1 cells expressing Mock, USP18 or USP18C64S treated with or without IFNα (1000U/ml) for 6 h. **n** Percentage of live WT, *USP18⁺/⁻* and *USP18⁻/⁻* 2fTGH cells after 48 h of treatment with or without IFNα (1000 U/ml). **o** Analysis of ICD markers in (**n**). **p** qRT-PCR of *IFIT1* and *PLK2* expression of STAT2 or STAT2 R148W expressing *STAT2⁻/⁻* (U6A) cells that were treated with or without IFNα (1000 U/ml) as indicated. **q** Percentage of live cells of STAT2 or STAT2 R148W expressing *STAT2⁻/⁻* (U6A) cells that were treated with or without IFNα (1000 U/ml) for 48 h. **r** Western blot of cell lysates in (**q**). **s** Analysis of IL-1β secretions of culture supernatant in (**q**). **t** Analysis of ICD markers in (**q**). *n* = 2 for Fig. 7p and *n* = 3 independent samples for (**b, c, e, f, h, l, k–o, q, s–t**) were analyzed. All data represent mean ± s.d, except where indicated. n.s. = not statistically significant. All *p*-value in Fig. 7 was determined by one-way ANOVA test. Source data are provided as a Source Data file.

its protein stability[43]. Since kinase activity does not contribute to the role of PLK2 in pyroptosis, we utilized the PLK2 kinase inhibitor TC-S7005 to stabilize PLK2 protein in AML cell lines and patient samples. PLK2 protein levels and cell death were both increased by TC-S7005 treatment in THP-1 and OCI-AML3 cells (Fig. 8i, j). Furthermore, TC-S7005 inhibited the growth of leukemia cells derived from two AML patients in a dose-dependent manner (Fig. 8k).

Since PLK2 induced GSDME dependent pyroptosis during *USP18* inhibition, we tested whether *Plk2* upregulation contributes to the survival benefit of mice transplanted with *Usp18* depleted leukemia cell in vivo. First, we generated doxycycline-inducible *Plk2* C1498 and AE9a leukemia cells to test whether expressing *Plk2* alone has any effect on leukemia development. We transplanted these cells into mice and observed significantly prolonged survival upon *Plk2* induction in vivo (Fig. 9a). Then, to examine the biological effect of *Plk2* upregulation during *Usp18* inhibition, we generated SgCtrl or SgPlk2 expressing *Usp18⁺/ᶠ UbcᴱᴿCre* AE9a cell lines. Cells were transplanted into recipient mice and Cre was activated after the mice developed AML. Again, *Usp18* depletion had a significant survival benefit and fewer cancer (GFP⁺) cells were observed in the peripheral blood (PB) of SgCtrl*Usp18⁺/ᐞ* mice compared to SgCtrl*Usp18⁺/ᶠ*. However, SgPlk2 *Usp18⁺/ᐞ* transplanted mice lost most of the survival benefit obtained from *Usp18* inhibition, indicating that *Plk2* induction contributes to leukemia cell death during *Usp18* depletion in vivo (Fig. 9b, c).

Altogether, we observed an expanded list of ISGs upon IFN treatment in *USP18* deficient cells and increased apoptotic/pyroptotic cell death. *USP18* depletion induces pyroptosis, which is dependent on the NLRP3 inflammasome-Caspase-1-GSDMD and Caspase-3-GSDME pathways. Among the atypical ISGs, we identified PLK2 as one mediator of the GSDME-dependent pyroptosis observed in a *USP18*-deficient context (Fig. 9d). Our study demonstrates that *USP18* deficiency expands the landscape of ISGs, facilitating immunogenic, pyroptotic cancer cell death in vivo.

## Discussion

Here, we showed that USP18, a major suppressor of IFN signaling, protects malignant cells from IFN-induced pyroptosis by suppressing the expression of canonical and non-canonical ISGs. Consistent with this finding, reduced expression of *USP18* is correlated with better survival in some human cancer subtypes and delays cancer progression in hematopoietic and solid tumor mouse models. We showed that type I IFN induces pyroptotic cell death in malignant cells when combined with the inhibition of endogenous USP18. Therefore, targeting *USP18* may expand the use of type I IFN treatment to previously unresponsive cancer subtypes, by switching from non-immunogenic to ICD.

In this study, we found that depletion of *Usp18* induced ICD and consequently delayed cancer progression in both hematological and solid malignancies. ICD was confirmed by the presence of conventional ICD indicators such as CRT exposure, and LDH release. In the leukemia models, we uncovered that Usp18 has protective role from

pyroptosis of leukemia cells including LSC-like population, essential for relapse. Although further investigation is needed to reveal the underlying mechanism of LSC specificity, one possible explanation is that hematopoietic stem cells (HSCs), from which LSCs arise, and LSCs in myeloproliferative neoplasms (MPN) are known to more sensitive to chronic IFN exposure[44,45]. Inhibition of USP18 not only enhances the type I IFN response, but also prolongs the IFN response, resulting in chronic IFN signaling. Importantly, our scRNA-seq data revealed that LSC-like cells expressed genes related to inflammasome activation/pyroptosis, supporting the possibility that ICD induced by Usp18 depletion promotes the elimination of cancer stem cells. Further dissection of the ICD phenotype in murine and human cells revealed that IFN treatment of *USP18* depleted cells induced (1) NLRP3 inflammasome-Caspase-1-GSDMD-mediated canonical pyroptosis and (2) Caspase-3-GSDME-mediated non-canonical pyroptosis. Interestingly, GSDMD deficiency did not affect the cell death of IFN-treated *USP18⁻/⁻* cells but reduced DAMPs and IL-1β release. This phenotype is not surprising because pyroptosis is not always deadly[46]. Conversely, GSDME deficiency suppressed the cell death and DAMPs release in IFN-treated *USP18⁻/⁻* cells.

Mechanistically, we discovered that IFN treatment of *USP18*-depleted cells resulted in considerable ISG expansion, including the upregulation of ISRE/NF-κB target genes. Although USP18 is located in both the cytoplasm and nucleus[47], only its cytoplasmic role inhibiting the IFN-receptor JAK-STAT signaling pathway has been described[11,18]. Here, we elucidated the role of nuclear USP18 in regulating the ISG enhancer landscape. Motivated by our previous finding that USP18 binds to STAT2[19], we discovered that nuclear USP18 inhibits the binding of ISGF3 complexes to both ISRE and IRE DNA motifs in a STAT2-dependent manner. We further revealed that USP18 can regulate the expression of previously unrecognized ISGs activated by IRF9/STAT2 in combination with NF-κB. Importantly, it was previously reported that IRF9/STAT2 can recruit p65 to NF-κB motifs in the vicinity of ISRE motifs to activate IL-6 expression[48], supporting this mechanistic finding. Overall, USP18 is a critical modulator of the 'ISG enhanceosome' complex[49], where each factor is essential for typical and atypical ISG transcription.

Considering the newly ascribed nuclear role of USP18 and the unique finding that USP18 depletion induced ICD when combined with IFN treatment, we hypothesized that targeting USP18 activates ICD-related atypical ISGs. Among atypical ISGs, we implicated PLK2 as one mediator of the observed cancer cell ICD. Enhanced levels of *PLK2* correlated with high levels of GSDME cleavage, a hallmark of pyroptosis[35,50], in IFN-treated *USP18⁻/⁻* THP-1 and MDA-MB-231 cancer cells. Importantly, we showed that ectopic expression of PLK2 in several types of cancer cells promoted processing of Caspase-3, cleavage of GSDME, DAMPs release, and cell death, but not IL-1β secretion. Since the NLRP3 inflammasome can be formed and activated by DAMPs, PLK2-induced DAMPs release could also be important for GSDMD pathway activation in USP18 depleted tumor environment in vivo. It will be interesting to examine how PLK2

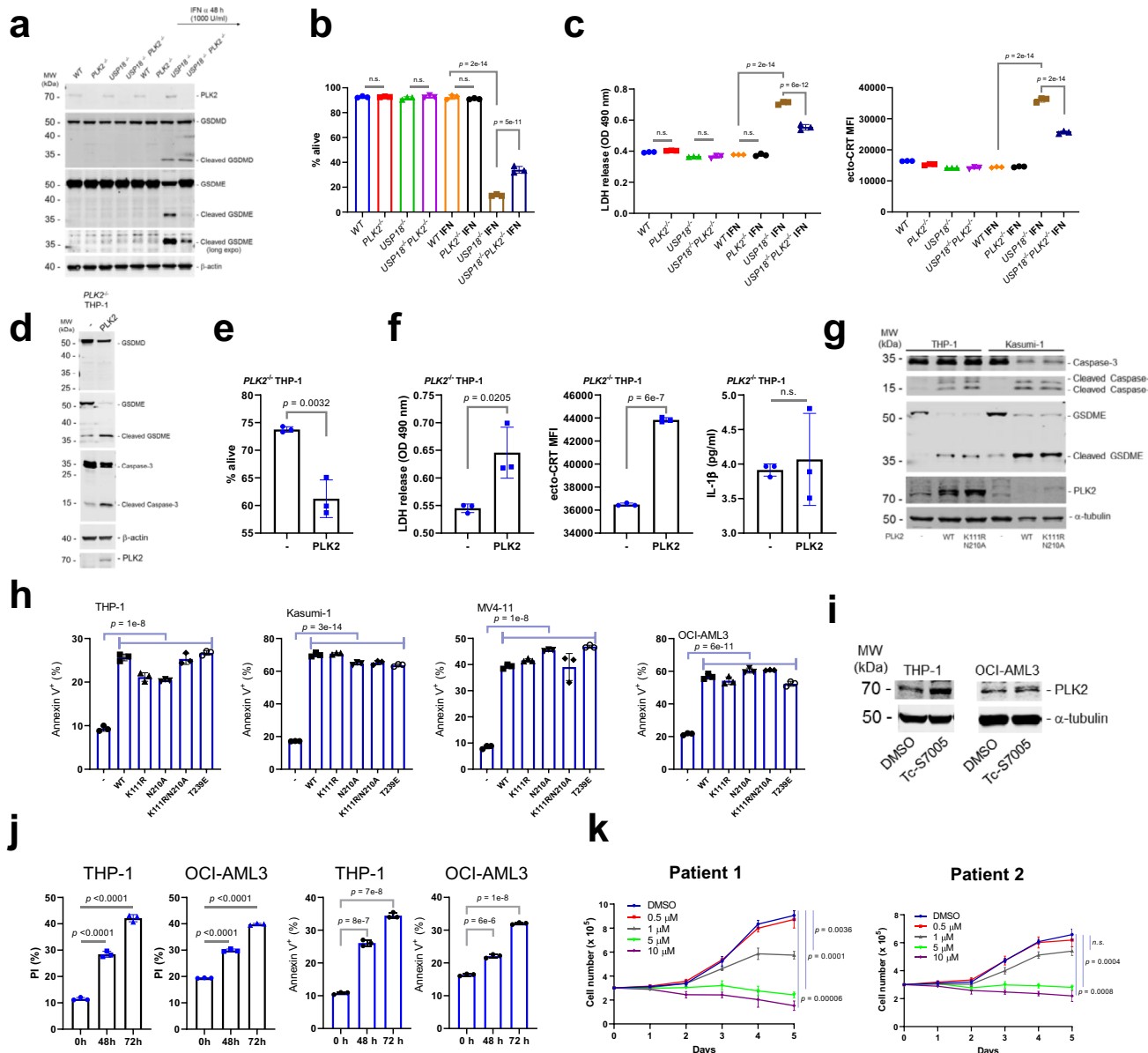

**Fig. 8 | USP18 loss-induced atypical ISG PLK2 regulates GSDME cleavage dependent pyroptosis. a** Western blot of WT, *PLK2⁻/⁻*, *USP18⁻/⁻*, and *USP18⁻/⁻PLK2⁻/⁻* THP-1 cells treated with or without IFNα (1000 U/ml) for 48 h with indicated antibodies. **b** Percentage of live cells in (**a**) (*n* = 3 independent samples each). *p*-value was determined by one-way ANOVA test. **c** Analysis of ICD markers (LDH release and ecto-CRT) in (**b**) (*n* = 3 independent samples each). *p*-value was determined by one-way ANOVA test. **d** Western blot of *PLK2⁻/⁻* THP-1 cells infected with Mock or PLK2 after two days of blasticidin selection using the indicated antibodies. e; Percentage of live cells in (**d**) (*n* = 3 independent samples each). *p*-value was determined by two-tailed Student *t*-test. **f** Analysis of ICD markers (LDH release and ecto-CRT) and IL-1β secretions in (**d**) (*n* = 3 independent samples each). *p*-value was determined by two-tailed Student *t*-test. **g**; THP-1 and Kasumi-1 cells were transduced with MIP, MIP-PLK2 WT, or MIP-PLK2 enzyme activity mutants. Two days after puromycin selection, cell lysates were analyzed by western blot with indicated

antibodies. **h** Ectopic PLK2 expression induces cell death in several AML cell lines independent of its kinase activity. THP-1, Kasumi-1, and MV4-11 cells were infected with MIP, MIP-PLK2 WT, or MIP-PLK2 enzyme activity mutants. Two days after puromycin selection, live cells were analyzed by FACS (*n* = 3 independent samples each). *p*-value was determined by one-way ANOVA test. **i** THP-1 or OCI-AML3 cells were treated with DMSO or TC-S7005 (final 10 μM) for 48 h, then cell lysates were analyzed. **j** THP-1 or OCI-AML3 cells were treated with DMSO or TC-S7005 for indicated times, then PI (%) and Annexin V⁺ (%) were analyzed by FACS (*n* = 3 independent samples each). *p*-value was determined by one-way ANOVA test. **k** Two different PDX AML cells were cultured with different dose of TC-S7005, and analyzed cell growth (*n* = 3 independent samples each conditions). All data represent mean ± s.d, except where indicated. n.s. not statistically significant. *p*-value was determined by one-way ANOVA test. Source data are provided as a Source Data file.

activates the Caspase-3-GSDME pathway in the future. Importantly, we also validated the biological effect of *Plk2* in WT and *Usp18*-depleted leukemia cells in vivo. Since AML levels (GFP) were restored by *Plk2* inhibition and the LSC-like population was most reduced by *Usp18* depletion in leukemic mice, we hypothesize that Plk2-mediated pyroptosis contributes to LSC reduction in *Usp18*-depleted leukemia. Suppression of PLK2 kinase activity enhances

the protein level of PLK2. Consequently, a PLK2 kinase inhibitor might have potential as a cancer therapeutic agent.

Altogether, our data supports the potential of targeting *Usp18* for cancer therapy. Importantly, we demonstrated a significant survival benefit by the heterozygous deletion of *Usp18*, which does not affect normal cells. Supporting the potential of a therapeutic window, interferonopathy has not been detected in humans or *Salmonella*

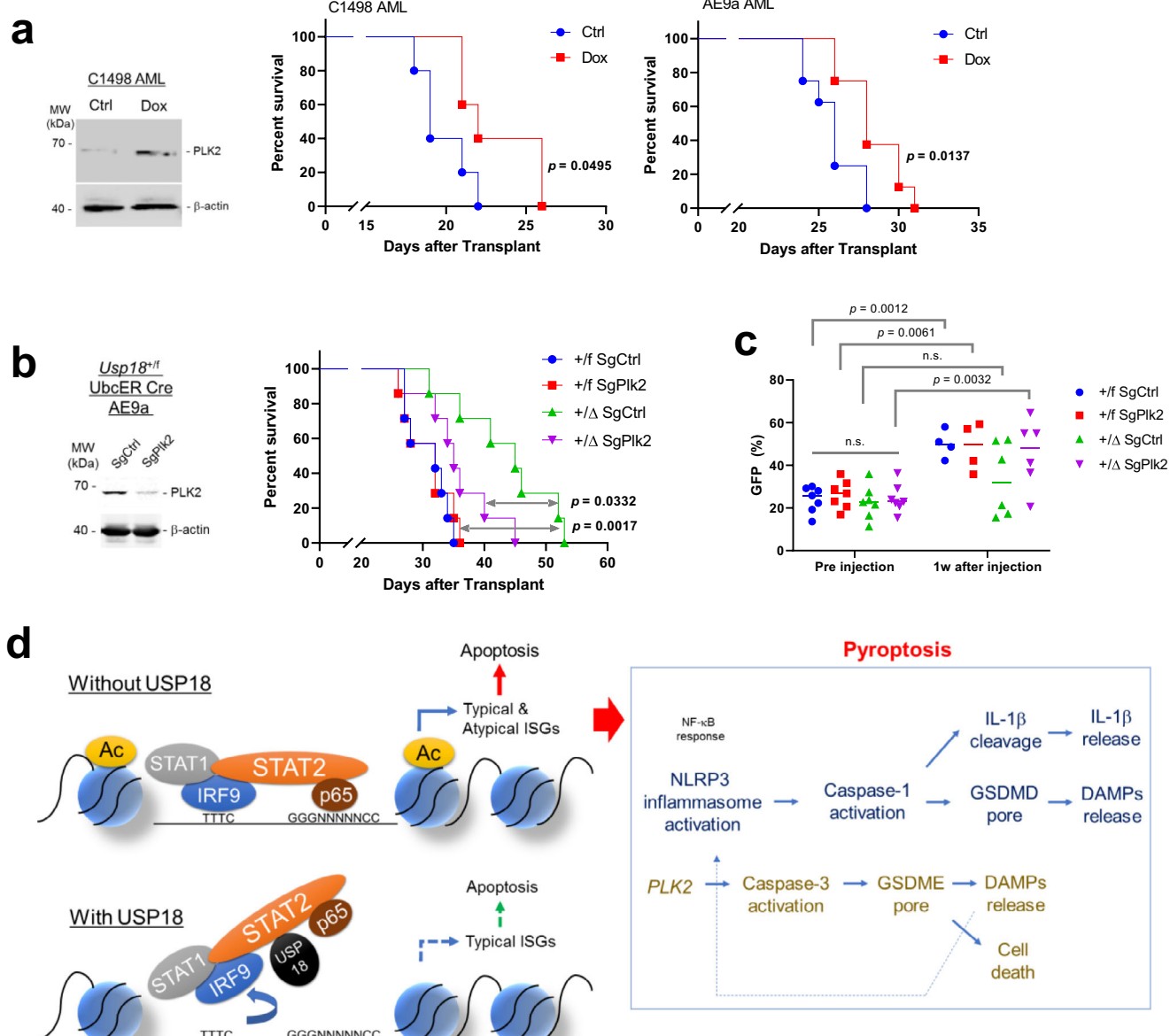

**Fig. 9 | USP18 loss-induced atypical ISG PLK2 regulates leukemogenesis.**
**a** Survival of sub-lethally irradiated recipient mice transplanted (i.v.) with *Plk2* doxycycline inducible C1498 or AE9a cells (C1498, *n* = 5 mice each, AE9a, *n* = 8 mice each). Two weeks after transplantation, mice were fed with or without doxycycline water (400 μg/ml, 5 % sucrose). *p*-value was determined by two-tailed Gehan–Breslow–Wilcoxon test (for C1498) or log-rank test (for AE9a). PLK2 induction in C1498 cells are also shown. **b** Survival for recipients of SgCtrl or SgPlk2 infected *Usp18^{+/f or +/Δ}* *UBC^{ER}-Cre* AE9a-GFP AMLs. *p*-value between +/f SgCtrl and +/Δ SgPLK2, and +/Δ SgCtrl and +/Δ SgPLK2 were determined by log-rank test. (right, *n* = 7 mice each group). Western blot of PLK2 protein level in SgRNAs expressing cells (left). **c** Analysis of GFP % of (m) in PB. *p*-value was determined by one-way ANOVA test. **d** Model of USP18-STAT2 mediated ISGs regulation followed by pyroptosis. Source data are provided as a Source Data file.

Typhimurium infected mice with heterozygous USP18 depletion[51–53]. Since tumors have higher STING activation and consequently more local type I IFN production compared to normal tissue[54], tumor sites should be more sensitive to targeting *USP18*. Furthermore, inducing ICD by targeting USP18 should prompt the release of DAMPs, resulting in more IFN production. Consequently, this therapeutic strategy would enhance IFN-dependent gene expression programs in multiple ways, resulting in a more robust anti-cancer response compared to other ICD induction methods. Based on our mechanistic data, an important next step is to develop a USP18 inhibitor that inhibits its binding with STAT2 but does not necessarily alter its enzymatic activity. One possible approach to generate a USP18 inhibitor is to perform a compound screen in THP-1 cells co-expressing USP18 and an ISRE reporter[55].

In summary, we discovered that USP18 plays an essential role in regulating cancer immunogenic, pyroptotic cell death. Thus, blocking USP18 represents a possible immunotherapeutic strategy to treat cancer.

## Methods

### Mice
All the animal studies were approved by the University of California San Diego (UCSD) Institutional Animal Care and Use Committee (IACUC, protocol # S07271); all experiments in this study adhere with all relevant ethical regulations for animal research.

Conditional *Usp18* knockout mice (*Usp18^{f/f}*) were generated in this study and crossed with *UBC^{ER}-Cre* mice (Stock 008085, B6.Cg-Tg-

*Ndor1*[(UBC-cre/ERT2)1Ejb]/2J; The Jackson Laboratory). C57BL/6 background *Usp18*[+/-] and *Ifnar1*[-/-] mice were previously described[56]. C57BL/6 WT mice were used as recipients for all non-competitive transplantation experiments. B6.SJL-*Ptprc*[a] *Pepc*[b]/BoyJ (CD45.1) (Stock 002014) mice from The Jackson Laboratory were used as recipients for the competitive transplantation experiment in supplementary Fig. 2o, p. The day of the first injection was defined as day 0. All mice in this study are both sex 8–12 weeks age were used. Animal vivarium condition; light/dark cycle: 12 h, temperature: 18–23 degrees C, humidity: 40–60%.

## Patient cohort analysis

*USP18* expression data from multiple cancer types and clinical annotations from human AML, ACC, and LGG patients were analyzed using GEPIA2[57]. The top and bottom quartiles were used to define groups for analysis. For breast and lung cancer patients, the relapse-free and overall survival correlation was assessed by KMplot[58], respectively. Patient cohorts were restricted to SEER prevalence. For lung cancer patients, GSE30219 was used.

## AML patient scRNA-seq analysis

scRNA-seq from three AML patients who had been given standard chemotherapy were analyzed (GSE116256). The average expression of *USP18* and *IFIT1* in malignant cells were quantified for patients AML329, AML556, and AML707B both before and after chemotherapy treatment.

## Mouse models of AML driven by AML1-ETO 9a and MLL-AF9

Mouse fetal liver (FL) cells derived from *Usp18*[+/f]*UBC*[ER]*-Cre* or wild-type mice were transduced with MSCV-IRES-GFP retroviruses expressing AML1-ETO9a (AE9a) or MLL-AF9 (MF9) and were transplanted intravenously (i.v.) into lethally irradiated (9 Gy) recipient mice. AE9a or MF9-expressing leukemia cells were harvested from spleens of moribund mice. AE9a leukemia cells were secondarily transplanted into sub-lethally irradiated (6 Gy) recipient mice.

## Mouse models of solid tumors

*Usp18*[+/+], *Usp18*[+/-], or *Usp18*[-/-] B16F10 cells ($1 \times 10^5$) or MC38 cells ($5 \times 10^5$) were subcutaneously (s.c.) injected into C57BL/6 mice. Tumor volumes were calculated over time: volume = length × width$^2$ × 0.5. The maximum tumor size/burden (2 cm × 2 cm$^2$ × 0.5 = 4000 mm$^3$) permitted by our institutional review board was not exceeded in our study.

## Gold standard vaccination assay

The same number of B16F10 *Usp18*[+/-] cells were treated with IFNβ (1000 U/ml) for 72 h or 20 μM cisplatin for 24 h. Both treatments resulted in 60–70 % cell death. After being washed with PBS, $3 \times 10^5$ treated cells were injected into the left flank of each C57BL/6 mouse (s.c. vaccination site). PBS was injected as the control. One week later, the vaccinated mice were challenged with a subcutaneous injection of $1 \times 10^5$ living B16F10 cells into their right flanks (challenge site). For the MC38 tumor vaccination assay, MC38 *Usp18*[+/+] or *Usp18*[+/-] cells were treated with or without IFNβ (1000 U/ml) for 72 h. After being washed with PBS, $1 \times 10^5$ treated cells were injected into the left flank of each C57BL/6 mouse (s.c. vaccination site). One week later, the vaccinated mice were challenged with a subcutaneous injection of $3 \times 10^5$ living MC38 cells into their right flanks (challenge site). Tumor incidence and growth in each group of mice were routinely monitored after the challenge. Tumor volumes were calculated: volume = length × width$^2$ × 0.5.

## Tamoxifen, anti-IFNAR1 antibody, doxycycline, and chemotherapy drug administration to mice

In all models of *Ubc*[ER]*-Cre* mediated *Usp18* depletion, tamoxifen (Sigma; T5648) was dissolved in corn oil (Sigma; C8267) and

intraperitoneally injected (i.p.) into transplanted mice (100 mg/kg). For the competitive bone marrow transplantation assay, mice were also fed a diet containing tamoxifen: 2020 Teklad Global Soy Protein-Free Rodent Diet 949.75 g, Sucrose 49.75 g, Tamoxifen 0.25 g, and Red Food color 0.25 g/Kg. This diet will provide ~40 mg tamoxifen per kg body weight, assuming 20–25 g body weight and 3–4 g intake. In the primary leukemia transplantation mouse models, oil or tamoxifen injection began three weeks after transplantation. To deplete *Usp18* in transplanted mice with established leukemia, mice were first allowed to become sick. Then mice were split into two groups with equal GFP% (AML) in their peripheral blood (PB) and injected with oil or tamoxifen. Control anti-mouse-IgG1 (0.25 mg/mouse, MOPC-21, #BP0083, BioX-Cell) or anti-mouse-IFNAR1 (0.25 mg/mouse, MAR1-5A3, #BE0241, BioXCell) was intravenously injected with the first oil or tamoxifen treatment in the corresponding experiments.

To evaluate the role of Plk2 in vivo, mice were first transplanted with either C1498 or AE9a AML cells harboring doxycycline inducible Plk2. Two weeks after transplantation, the mice were given water with or without doxycycline (Sigma; D9891) (400 μg/ml, 5% sucrose).

For experiments assessing the combinatorial effect of chemotherapy and *Usp18* depletion, chemotherapy treatment was initiated in mice upon detection of >5% GFP$^+$ cells in their PB by flow cytometry. Mice were treated every 24 h for five consecutive days with i.p injections of 100 mg/kg cytarabine (Hospira, Zydus Hospira Oncology Private Ltd, India). During the first 3 days, 3 mg/kg doxorubicin (APP, Fresenius Kabi USA) was also administered via a separate i.p. injection. Mice treated with chemotherapy were given ciprofloxacin-supplemented drinking water (ciprofloxacin (Sigma; #17850) 125 mg/L, sucrose (Sigma; S9378) 20 g/L) until 2 weeks after the end of chemotherapy treatment. Immediate response and long-term treatment effects were monitored weekly by flow cytometry and blood counts.

## AE9a cell lines

To generate AE9a cell lines, primary spleen cells from *Usp18*[+/+] AE9a-GFP or *Usp18*[+/f] *Ubc*[ER]*-Cre* AE9a-GFP secondary transplantation recipient mice were grown in either IMDM media supplemented with 20% FBS and both IL3 and SCF conditioned media (for *Usp18*[+/+] AE9a-GFP cell line) or phenol red free IMDM media supplemented with 20% charcoal stripped FBS (Gibco; A3382101) and 10 ng/ml each of IL-3 (#213-13; Peprotech) and SCF3 (#250-03; Peprotech) (for *Usp18*[+/f] *Ubc*[ER]*-Cre* AE9a-GFP cell line) for several weeks. Cells were able to be maintained without added cytokines after several weeks in culture.

## Plasmid construction

Human *USP18*, human *PLK2*, and *PLK2* kinase domain mutant cDNAs were cloned into the MSCV-IRES-Puro (MIP) retroviral vector. SgRNA resistant human *USP18*, *USP18* C64S mutant and *PLK2* cDNAs were cloned into the pCX4-bsr retroviral vector. ShRNA plasmids (pLKO.1 vectors) targeting human PLK2 (TRCN0000000868: ShPLK2-2, TRCN0000000869: ShPLK2-3) and a control shRNA were purchased from La Jolla Institute (LJI). SgRNA for Caspase-1 (target sequence; CACACGTCTTGCTCTCATTA) and murine Plk2, (target sequence; ACGGGGAAGCGCTACTGCCG) were cloned into lentiCrispr V2 plasmid. For doxycycline mediated *Plk2* induction, murine *Plk2* cDNA was cloned into the doxycycline-inducible lentiviral vector (pTRIPZ backbone from Addgene #83468). Expression constructs for *IRF9*, *STAT1*, and *STAT2* were previously created as described[19]. The following luciferase reporter plasmids were created by cloning into the pGL3-basic plasmid: ISRE-Luc, IRE-Luc, *PLK2* enhancer-Luc, *PLK2* enhancer ΔISRE-Luc, *CCL4L2* promoter-Luc, and *CCL4L2* promoter ΔNF-κB-Luc.

## Transfection and viral infection

Transfection was conducted using PEI (Polyethylenimine). For retrovirus production, 293T cells were co-transfected with plasmids

encoding viral vectors and packaging vectors: pCL-10A1 for infection of human cells or Ecopac for murine cells. For lentivirus production, gag-, pol-, and env-expressing packaging plasmids (pMD2.G: Addgene #12259 and psPAX2: Addgene #12260) were used. Viral particles were collected 48 h after transfection and passed through a 0.45 µm sterile filter. For retroviral or lentiviral infection, cells were spin infected (2000 × g, 3 h, 30 °C; Allegra X12R [Beckman Coulter]) with polybrene (8 µg/ml).

## Cell culture

293T (ATCC), MDA-MB-231 (ATCC) (kindly provided by Dr. David Cheresh, previously), U5A (IFNAR2$^{-/-}$), and U6A (STAT2$^{-/-}$) (kindly provided by Dr. George Stark, previously) cells were grown in DMEM media supplemented with L-glutamine, penicillin/streptomycin, and 10% FBS. THP-1 (ATCC), MV4-11 (ATCC), MOLM13 (ATCC), OCI-AML3 (ATCC), Kasumi-1 (ATCC), B16F10 (ATCC) and MC38 (NCI) cells were grown in RPMI media supplemented with glutamine, penicillin/streptomycin, and 10% FBS. For colony assay, cells were grown (2 × 10$^5$ cells per plate) in M3434 medium (STEMCELL Technologies) G; Granulocyte, GM; Granulocyte/Macrophage, GEMM; Granulocyte/Erythroid/Macrophage/Megakaryocyte, M; Macrophage, E; Erythroid. Primary bone marrow (BM) or fetal liver cells from *Usp18* conditional mice were maintained in IMDM media supplemented with 20% FBS and both IL3 and SCF conditioned media.

## Crispr knockout cell lines

CRISPR gene editing technology was used to delete USP18 in THP-1, MDA-MB-231 and 2fTGH, murine Usp18 in B16F10 and MC38, and PLK2, GSDMD, and GSDME in THP-1 and *USP18*$^{-/-}$ THP-1 cells. Guide RNA sequences were cloned into the pSpCas9(BB)−2A-Puro (PX459) plasmid (Addgene #48139) or the lentiCrispr V2 puro plasmid (Addgene #52961). Virus preparation and infection were performed as previously described. Single cells were then sorted into 96-well plates and incubated without selection. Individual clones were screened for genes of interest knockouts by PCR, DNA sequencing, and Western blot. Guide RNA sequences and primers are provided in Supplementary Table 4.

## Western blotting and antibodies

Cells were lysed in mild RIPA buffer composed of 25 mM Tris-HCl (pH 8.0), 150 mM NaCl, 1 mM EDTA, and 0.5% Nonidet P-40 (NP-40). Proteinase and phosphatase inhibitors (Roche) were added to all lysis buffers. Cell lysates were centrifuged (20,000 × g) at 4 °C for 5 min. All samples were then denatured in 1× sample buffer (50 mM Tris-HCl [pH 6.8], 2% SDS, 2-mercaptethanol, 10% glycerol, and 1% bromophenol blue) for 5 min at 100 °C. For the detection of cleaved Caspase-1 and cleaved GSDMD, cell culture supernatants were incubated with Blue Sepharose 6 Fast Flow (GE Healthcare; GE17-0948-01) for 2 h, then precipitated with three volumes of cold methanol. Antibodies used include: FLAG (M2; Sigma), Myc (9E10; Santacruz), HA (3F10; Roche), α-tubulin, β-actin, STAT1 (#9172; CST), phospho-STAT1 (#9167; CST), PLK2 (#14812; CST), H2AX (#2595; CST), γH2AX (#9718; CST), LC3B (#3868; CST), AIM2 (#12948; CST), GSDMD (#96458, or #39754; CST), GSDME (ab215191; Abcam), Caspase1 (sc-56036; Santacruz), Cleaved Caspase1 (D57A2, #4199; CST), Cleaved IL-1β (D3A3Z, #83186; CST), Caspase3 (#9662; CST), p84 (5E10, GTX70220; GeneTex), GAPDH (GT239, GTX627408; GeneTex) and NF-κB p65 (#8242; CST). For anti-murine PLK2 detection, Anti-snk (E-10) (PLK2) antibody (sc-374643; Santacruz) was used. Anti-murine ISG15, human USP18, and murine USP18 antibodies were used as previously described[19,56]. All antibodies for western blot analysis were used at 1:1000 dilution. The LI-COR Odyssey system was used for image acquisition and quantification.

## DNA-pulldown assay

293T or U6A cells were transfected with plasmids encoding FLAG-tagged: empty vector, IRF9, STAT2, or USP18. Cell lysates were incubated with M2 FLAG-beads (Sigma). FLAG-tagged proteins were eluted using 3× FLAG peptide.

3′ biotinylated ISRE, IRE, or unmodified DNA oligos were incubated with the indicated purified Flag-tagged proteins for 2 h at 4 °C in binding buffer (10 mM Tris, 1 mM KCl, 1% NP-40, 1 mM EDTA, 5% glycerol). 10% input was collected and streptavidin beads (RayBiotech; #65305) were added for 2 additional hours of incubation at 4 °C. Samples were washed three times in wash buffer (25 mM Tris·HCl [pH 8.0], 300 mM NaCl, 1 mM DTT, 1 mM EDTA, 0.1% SDS, 1% NP-40, and 0.5% sodium deoxycholate) and ISRE- or IRE-binding proteins were eluted by boiling and analyzed by immunoblotting.

## ELISA

Culture media was collected and analyzed for human IL-1β production using the Human IL-1β Quantikine enzyme-linked immunosorbent assay (ELISA) kit (R&D systems; DLB50) or Human interleukin 1 beta ELISA kit (MyBioSource.com; MBS263843). For the murine IL-1β quantification analysis in culture media, Mouse interleukin 1 beta ELISA kit (MyBioSource.com; MBS175967) was used.

## Luciferase assays

Luciferase assays were performed using the Dual-luciferase reporter assay system (Promega; E1910). Briefly, cells were transfected with IRF9, STAT2, NF-κB p65, pRL-TK Renilla luciferase plasmid (Promega), and the indicated pGL3 Firefly luciferase reporter plasmids. Empty pcDNA3.1 plasmid was added to equalize DNA amounts between wells. 24 h post-transfection, cells were lysed in passive lysis buffer (Promega; E1941). Firefly and Renilla luciferase activities were measured using the Dual-Luciferase Assay Kit (Promega; E1910) and a FLUOstar OPTIMA luminometer (BMG Labtech, Ortenberg, Germany), according to the manufacturer's instructions. Firefly luciferase activity was normalized to Renilla luciferase activity and the fold induction of each reporter was calculated by normalizing to activity of control cells transfected with empty vector.

## RNA isolation and qRT-PCR analysis

RNA was extracted with Trizol reagent (Thermo Fisher Scientific; 15596026). Reverse transcription (RT) reactions using equal amounts of RNA were performed using the qScript cDNA synthesis kit (Quanta Biosciences; #95048). The resulting cDNA templates were then analyzed by qRT-PCR using the KAPA SYBR FAST universal qPCR kit (Kapa Biosystems, Inc.) and a CFX96 thermal cycler (BIO-RAD). Primer sequences are as provided in Supplementary Table 5.

## PDX of Patient AMLs and growth assay with TC-S7005

All patient AML samples were collected with proper, informed consent and all experiments were performed according to an institutional review board-approved protocol, in accordance with the Declaration of Helsinki, and with an approved animal study IACUC protocol at Cincinnati Children's Hospital Medical Center (CCHMC). Residual diagnostic specimens from AML patients at CCHMC were treated with an OKT3 antibody and engrafted into NSGS mice. Sensitivity of these PDX-derived AML cells was assessed in StemSpan media (STEMCELL Technologies; #09600) supplemented with 50 ng/mL human SCF, TPO, FLT3L, IL-3, and IL-6 with/without TC-S7005. The PDX-derived AML cells were plated in triplicate with the following doses of TC-S7005 (TOCRIS): 0 (DMSO), 0.5, 1, 5, and 10 µM.

## Lactate dehydrogenase release assay

The lactate dehydrogenase (LDH) released from cells into culture supernatants was measured using the LDH Cytox™ Assay Kit (426401; BioLegend) according to the manufacturer's protocol. The percentage

of LDH release (Cytotoxicity) was calculated as follows: % Cytotoxicity = (OD490: interferon-treated LDH activity − spontaneous LDH activity) / (OD490: maximum LDH activity - spontaneous LDH activity) × 100.

## Flow cytometry

Primary cells (FL, PB, and BM cells) from mice were treated with ACK buffer at room temperature for 5 min. For stem cell analysis of BM, the lineage cocktail consisted of PerCP-Cy5.5 conjugated CD3 (17A2, #100218), CD4 (GK1.5, #100434), CD8a (53-6.7, #100734), CD11b (M1/70, #101228), Gr1 (RB6-8C5, #108428), B220 (RA3-6B2, #103236), CD19 (6D5, #115534), and Ter119 (TER-119, #116228), Sca-1-APC (E13-161.7, #122512), c-Kit–PE-Cy7 (2B8, #105814), CD48-PE (HM48.1, #103405), and CD150-biotin (TC15-12F12.2, #115908) (All antibodies are from Biolegend, except for CD48 (HM48.1) from eBioscience) together with streptavidin-APC-Alexa Fluor 750 (Invitrogen, SA1027) was used. LSK (Lin⁻ c-Kit⁺ Sca-1⁺) were further analyzed with CD48 and CD150 to define long-term HSC (LT-HSC), short-term HSC (ST-HSC), and multipotent progenitors (MPP). For stem cell analysis of FL, PerCP-Cy5.5 conjugated lineage cocktail (Biolegend), c-Kit–PE-Cy7 (2B8, #105814), and AA4.1 (CD93) -APC (#136510) (Biolegend) was used. For analysis or sorting LK (Lin⁻ c-Kit⁺) of splenocytes from leukemic mice, the following antibodies were used: c-Kit-PE-Cy7 (2B8, #105814) and PerCP-Cy5.5 conjugated lineage cocktail (Biolegend). For ecto-CRT staining, PE conjugated Calreticulin antibody (Cell signaling, clone D3E6, #19780) was used. Apoptotic cells were stained with Annexin V-APC (BDB550474), using the Annexin V apoptosis detection kit (BD) according to the manufacturer's protocol. Dead cells were stained with Propidium Iodide (PI) (Thermo, P1304MP). Flow cytometric analysis was performed using a BD FACSCanto with standard lasers and optical filters. Cell sorting was performed using a BD FACSAria-II. For immune cell profiling in Fig. 1e and Supplementary Fig. 10d, we used the following antibodies and analyzed samples with NovoCyte Advanteon (Agilent Technologies). Zombie Near-IR (#423108) and TruStain FcX™ (#101320) from BioLegend were used for viability staining and Fc block, respectively. Antibodies: CD45.2 (104, #109822) CD3 (17A2, #100241), CD4 (RM4-5, #100542), CD8a (53-6.7, #100710), CD25 (PC61, #102008), CD69 (H1.2F3, #104541), CD71 (RI7217, #113813), CD11b (M1/70, #101257), CD11c (N418, #117336), F4/80 (BM8, #123137), Gr-1 (RB6-8C5, #108452), NK1.1 (PK136, #108737), and CD19 (6D5, #115541) from BioLegend, CD19 (1D3, #553785) from BD, Foxp3 (FJK-16s, #17-5773-82) from eBioscience. All antibodies for FACS analysis were used at 1:100 dilution. Gating strategies in this study are provided in Supplementary Fig.12.

## Immunohistochemistry

MC38 tumors were harvested on Day 14, fixed in 4% paraformaldehyde, embedded in paraffin, and sectioned. After deparaffinization and antigen retrieval in citrate buffer, slides were blocked with BLOXALL Blocking Solution (Vector Laboratories SP-6000-100) and 2.5% normal horse or goat serum (Vector Laboratories) and incubated with the following primary antibodies overnight at 4 °C: anti-HMGB1 (rabbit monoclonal, clone D3E5, Cell Signaling Technology #6893, 1:100 dilution), anti-CD8a (rat monoclonal, clone 4SM16, eBioscience 14-0195-82, 1:100 dilution). For HRP-conjugated secondary antibodies and DAB substrate, ImmPRESS Horse Anti-Rabbit IgG PLUS Polymer Kit Peroxidase (Vector Laboratories MP-7801) and ImmPRESS Goat Anti-Rat IgG Polymer Kit Peroxidase (Vector Laboratories MP-7444) were used according to the manufacturer's instructions. Counterstain was performed with hematoxylin (Vector Laboratories H-3502). Slides were then dehydrated and mounted in Permount Mounting Medium (Fisher SP15-100). Images were captured with a BZ-X710 (Keyence) microscope.

## Proximity-based biotin labeling (BioID) and protein identification by mass spectrometry

U5A cells stably expressing USP18-BioID2 or Control-BioID2 were cultured in media (DMEM, 10% FBS, 1% Pen/Strep) supplemented with 50 µM biotin for 48 h. Approximately 20 million cells per sample were washed, lysed, and subjected to streptavidin-based co-immunoprecipitation to identify proximal proteins using the BioID2 method described in Roux et al.[59]. Mass spectrometry was performed by the Sanford Burnham Prebys Proteomics Shared Resource.

## scRNA-seq and analysis

GFP⁺ Lin⁻ c-Kit⁺ splenocytes were sorted by FACS from transplanted *Usp18^{+/f or f/f} UBC^{ER}-Cre* AE9a recipient mice following two injections with oil or tamoxifen. 8000 cells from each group (GFP⁺Lin⁻c-Kit⁺ *Usp18^{+/f}* and *Usp18^{+/Δ}* leukemia cells or GFP⁺Lin⁻c-Kit⁺ *Usp18^{f/f}* and *Usp18^{Δ/Δ}* leukemia cells) were coupled to beads using the 10× Genomics Chromium controller. Single cell sequencing libraries were prepared using the 10× Genomics chromium single cell 3′ reagent V3 kit according to the manufacturer's protocol and sequenced using an Illumina Hiseq 4000. Data were mapped using Cell Ranger version 3.0.2 (10× Genomics) to mm10 and analyzed using the Seurat R for single cell genomics[60].

CellHarmony was used to generate identities for all cells in an unsupervised fashion[61]. Cell types were applied to each cell based on the statistical correlation (Pearson coefficient > 0.4) between their gene expression profiles and those of published Lin⁻ c-Kit⁺ CD34⁺ progenitor reference profiles[62]. The standard workflow was followed as described in the workflow's associated documentation (http://www.altanalyze.org/cellHarmony/).

## Bulk RNA-seq and analysis

For transcriptomic analysis of AE9a leukemia cells with *Usp18* depletion, two sets of GFP⁺ Lin⁻ c-Kit⁺ splenocytes were sorted after administration of three oil or tamoxifen injections to transplanted *Usp18^{+/f} UBC^{ER}-Cre* AE9a mice (*n* = 3 mice each, one set used for paired ATAC-seq). RNA was extracted from sorted cells using Trizol reagent. For transcriptomic analysis of cell lines, RNA was extracted from untreated or IFNα (1000U/ml) treated (6 h) THP-1 and MDA-MB-231 WT and *USP18^{-/-}* cells. All RNA-seq libraries were prepared by Novogene and sequenced using an Illumina Novaseq 6000 (PE150). The data were analyzed using the RNASTAR/DeSeq2 pipeline.

## ATAC-seq and analysis

Permeabilized nuclei were obtained by resuspending cells in 250 µl Nuclear Permeabilization Buffer (10 mM Tris-HCL [pH 7.5], 10 mM NaCl, 3 mM MgCl₂, 0.1% Tween-20 [Sigma], 0.1% IGEPAL-CA630 [Sigma] and 0.01% Digitonin [Promega] in water) and incubating for 10 min on a rotator at 4 °C. Nuclei were then pelleted by centrifugation for 5 min at 500 × *g* at 4 °C. The pellet was resuspended in 20 µl ice-cold Tagmentation Buffer (33 mM Tris-acetate [pH 7.8] [BP-152, Thermo Fisher Scientific], 66 mM K-acetate [P5708, Sigma], 11 mM Mg-acetate [M2545, Sigma], 16% DMF [DX1730, EMD Millipore] in Molecular biology grade water [46000-CM, Corning]). An aliquot was then counted to determine nuclei concentration. Approximately 50,000 nuclei were resuspended in 10 µl ice-cold Tagmentation Buffer and incubated with 1 µl Tagmentation enzyme (FC-121-1030; Illumina) at 37 °C for 30 min. The tagmented DNA was purified using the MinElute PCR purification kit (28004, Qiagen). The libraries were amplified using NEBNext High-Fidelity 2× PCR Master Mix (M0541, NEB) with primer extension at 72 °C for 5 min and denaturation at 98 °C for 30 s, followed by 8 cycles of denaturation at 98 °C for 10 s, annealing at 63 °C for 30 s, and extension at 72 °C for 60 s. Amplified libraries were then purified using the MinElute PCR purification kit (28004, Qiagen) and two size selection steps were performed using SPRIselect bead (B23317, Beckman Coulter) with 0.55× and 1.5× bead-to-sample volume ratios,

respectively. Libraries were sequenced on an Illumina HiSeq 4000 to a total read depth ≥ 35 Million Read Pairs.

FASTQ files from ATAC-seq experiments were mapped to the human hg38 or mm10 genome using Bowtie2 (PMCID: PMC3436841) using default parameters. HOMER software (PMCID: PMC2898526) was used for further analysis. First, peaks were identified using findPeaks with the selected options: 'style factor -minDist 200 -size 200'. Next, peaks were merged using mergePeaks and signals were quantified using annotatePeaks.pl for each data replicate. Peaks within 3000 bp of the transcription start site were excluded, then differentially accessible regions were identified using DeSeq2 (PMCID: PMC4302049) through HOMER getDiffExpression.pl default parameters (fold change >2 and adjusted *p*-value <0.05). We used the default parameters of findMotifsGenome.pl to assess enrichment of known DNA motifs within the differential regions increased by IFN treatment and/or *USP18^−/−*.

## H3K27ac ChIP-seq and analysis

Untreated and IFN treated cells were crosslinked by adding 1% paraformaldehyde to the culture media and incubating at room temperature for 10 min. The crosslinked cells were quenched by adding ice-cold glycine to a final concentration of 125 mM and incubated for 5 min. Cells were then washed with PBS, resuspended in Supplement Cell Lysis buffer (5 mM PIPES [pH 8.0], 85 mM KCl, 0.5% NP-40) supplemented with 1 × proteinase inhibitor cocktail (Roche), and incubated on ice for 10 min. Nuclear pellets were lysed with SDS lysis buffer (1% SDS, 10 mM EDTA, 50 mM Tris-HCl, pH 8.1) containing 1 × proteinase inhibitor cocktail for 15 min prior to sonication. Cells were sonicated using the Bioruptor plus (Diagenode) on the High setting: 30 s On, 30 s Off for 20 cycles. SDS was removed from the chromatin solution by centrifugation for 10 min at 4 °C. The sheared chromatin was measured, diluted with ChIP Dilution buffer (0.01% SDS, 1.1% Triton X-100, 1.2 mM EDTA, 16.7 mM Tris-HCl, pH 8.1, 167 mM NaCl), and incubated with Protein A agarose beads for 1 h at 4 °C. An aliquot was removed as input from the pre-cleared chromatin and 300 μg of each sample was then incubated with H3K27ac antibody (Active Motif #91193) overnight at 4 °C. Then, Protein A agarose beads were added for 1 h at 4 °C. The immune complexes were washed once with low salt wash buffer (0.1% SDS, 1% Triton X-100, 2 mM EDTA, 20 mM Tris-HCl, pH 8.1, 150 mM NaCl), once with high salt wash buffer (0.1% SDS, 1% Triton X-100, 2 mM EDTA, 20 mM Tris-HCl, pH 8.1, 500 mM NaCl), once with LiCl wash buffer (0.25 M LiCl, 1% NP-40, 1% deoxycholate, 1 mM EDTA, 10 mM Tris-HCl, pH 8.1), and twice with 1× TE buffer (10 mM Tris-HCl, 1 mM EDTA [pH 8.0]). The complexes were eluted with elution buffer (1% SDS, 0.1 M NaHCO₃). The eluates (200 μl) were reverse crosslinked at 65 °C overnight by adding 8 μl of 5 M NaCl, followed by RNase A treatment for one hour at 37 °C. The samples were mixed with 4 μl of 0.5 M EDTA, 8 μl of 1 M Tris-HCl pH 8.0, and 1 μl of Proteinase K for two hours at 45 °C. The ChIPed DNA was purified using ChIP DNA clean & Concentrator (Zymo Research; D5205). The libraries for ChIP-seq were prepared by Novogene and sequenced on an Illumina Novaseq 6000 (PE150).

The FASTQ files from ChIP-seq experiments were mapped to the human hg38 genome using Bowtie2 with default parameters. HOMER software was used for further analysis. First, pooled ATAC-seq peaks were sorted into promoter proximal or promoter distal peaks using a cutoff of within 3000 bp of the transcription start site. Then H3K27ac ChIP-seq data was quantified at each peak using annotatePeaks.pl within a 1000 bp window of the ATAC-seq peak, as described previously (PMCID: PMC7305990). Differentially acetylated regions were identified using DeSeq2 through HOMER getDiffExpression.pl default parameters (fold change > 2 and adjusted *p*-value <0.05). Next, peak windows were shifted back to the original size of the ATAC-seq data and findMotifsGenome.pl was used to assess enrichment of known

DNA motifs within the differential regions increased by IFN treatment and/or *USP18^−/−*.

## Sequence data visualization

Bulk RNA-seq, ATAC-seq, and H3K27ac ChIP-seq data were visualized using the UCSC Genome Browser (PMCID: PMC186604) or through custom R scripts.

## ChIP-qPCR assay

Cells were crosslinked by adding 1% paraformaldehyde to the culture media and incubating at room temperature for 10 min. The crosslinked cells were quenched by adding ice-cold glycine to a final concentration of 125 mM and incubating for 5 min. Cells were then washed with PBS, resuspended in Supplement Cell Lysis buffer (5 mM PIPES pH 8.0, 85 mM KCl, 0.5% NP-40) supplemented with 1 × proteinase inhibitor cocktail (Roche), and incubated on ice for 10 min. Nuclear pellets were lysed with SDS lysis buffer (1% SDS, 10 mM EDTA, 50 mM Tris-HCl, pH 8.1) containing the 1 × proteinase inhibitor cocktail for 15 min prior to sonication. Cells were sonicated in 10 s pulses of 30% amplitude, 6 times (10 times for the mouse splenocytes) with 35 s interval. SDS was removed from the chromatin solution by centrifugation for 10 min at 4 °C. The sheared chromatin was measured, diluted with ChIP Dilution buffer (0.01% SDS, 1.1% Triton X-100, 1.2 mM EDTA, 16.7 mM Tris-HCl, pH 8.1, 167 mM NaCl), and incubated with Protein A agarose beads for 1 h at 4 °C. An aliquot was removed as input from the pre-cleared chromatin and 50 μg of each sample was incubated with control-IgG, NF-κB p65 (D14E12; Cell Signaling), IRF9 (D2T8M; Cell Signaling), or H3K27ac (#39133; Active Motif, for mouse splenocytes) antibody overnight at 4 °C. Then, Protein A agarose beads were added for 1 h at 4 °C. The immune complexes were washed once with low salt wash buffer (0.1% SDS, 1% Triton X-100, 2 mM EDTA, 20 mM Tris-HCl, pH 8.1, 150 mM NaCl), once with high salt wash buffer (0.1% SDS, 1% Triton X-100, 2 mM EDTA, 20 mM Tris-HCl, pH 8.1, 500 mM NaCl), once with LiCl wash buffer (0.25 M LiCl, 1% NP-40, 1% deoxycholate, 1 mM EDTA, 10 mM Tris-HCl, pH 8.1), and twice with 1x TE buffer (10 mM Tris-HCl, 1 mM EDTA [pH 8.0]). The complexes were eluted with elution buffer (1% SDS, 0.1 M NaHCO₃). The eluates (200 μl) were reverse crosslinked at 65 °C overnight by adding 8 μl of 5 M NaCl, followed by RNase A treatment for one hour at 37 °C. The samples were mixed with 4 μl of 0.5 M EDTA, 8 μl of 1 M Tris-HCl pH 8.0, and 1 μl of Proteinase K for 2 h at 45 °C. The ChIP DNA was purified using the QIAquick PCR purification kit (Qiagen; #28104). Primers used for ChIP-qPCR analysis are listed in Supplementary Table 6.

## Gene set enrichment analysis

Gene set enrichment analysis[63] was performed using the online GSEA tool (https://www.gsea-msigdb.org/gsea/index.jsp).

## ChIP-Atlas enrichment analysis

Promoters/enhancers of typical and atypical ISGs were analyzed by Chip-Atlas enrichment analysis (https://chip-atlas.org/enrichment_analysis).

## Ingenuity pathway analysis

Upstream regulators of typical, atypical, and non-canonical ISGs were analyzed by Qiagen Ingenuity pathway analysis (IPA) (https://digitalinsights.qiagen.com/products-overview/discovery-insights-portfolio/analysis-and-visualization/qiagen-ipa/).

## Reactome analysis

Pathway analysis for several gene sets in this study were performed using the reactome pathway online tool (https://reactome.org/).

## Gene ontology (GO) analysis

GO analysis for several gene sets in this study were performed using the GO online tool (http://geneontology.org/).

## Interferome analysis

Interferome analysis[64] was performed using the online Interferome tool v2.01 (http://www.interferome.org/interferome/home.jspx).

## Statistical analysis and reproducibility

Data were collected and analyzed in either GraphPad 8 or Microsoft excel. Statistical analysis; Comparison of results from two groups was done by two-tailed Student's *t* test. For multiple comparisons, results were analyzed by one-way or two-way ANOVA. For multiple comparisons, adjustments were made. Survival data were presented as Kaplan–Meier curves and a log-rank test or Gehan-Breslow-Wilcoxon test was performed. All exact *p*-value are indicated in figures and source data files except for Supplementary Fig. 1b. *p*-values for all multiple comparisons, IPA, RNA-seq, HOMER, Reactome, GO and ChIP enrichment tool analysis were corrected for multiple testing using Benjamini–Hochberg false discovery rate approach. A *p*-value of <0.05 was considered significant. Data for Supplementary Figs. 9a, 10b are represented as means ± standard errors (SEM). All other results are represented as means ± standard deviations (SD) unless otherwise stated. All western blot, qPCR and FACS experiments were performed in at least two biologically independent samples and reproducibly confirmed at least twice. All mice experiments, when feasible, were repeated at least twice and reproducibly confirmed. We also provided the data for another CRISPR KO clone in source data file for critical experiments presented in Figs. 6d, 7b, c, n, o, 8b, c and Supplementary Fig. 9b. RNA-seq for THP-1 and MDA-MB-231 were run in biologically triplicates. ATAC-seq was run in biologically duplicates.

## Reporting summary

Further information on research design is available in the Nature Portfolio Reporting Summary linked to this article.

## Data availability

Source data are provided with this paper. All data reported in this work are available within the Article, Supplementary information and Source Data file. Original data for RNA-seq, ATAC-seq, ChIP-seq, and scRNA-seq have been deposited in the Gene Expression Omnibus (GEO) database under accessions GSE165424-GSE165429 and GSE196580-GSE196582. GSE165424; GSE165425; GSE165426; GSE165427; GSE165428; GSE165429; GSE196580; GSE196581; GSE196582. Source data are provided with this paper.

## Code availability

We used Seurat Version 3 for analyzing scRNA-seq data. Code used for generating all figures related to scRNA-seq are available in GitHub repository. For heterozygous Usp18 depletion; [https://github.com/kei16-cell/AE9a-scRNAseq-Usp18-heterozygous-deletion] . For homozygous Usp18 depletion; [https://github.com/kei16-cell/AE9a-scRNAseq-Usp18-homozygous-deletion].

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

## Acknowledgements

ATAC libraries were generated at the UC San Diego Center for Epigenomics, which was supported in part by the UC San Diego School of Medicine. UCSD CCBB for cell ranger analysis was supported by Altman Clinical and Translational Research Institute (ACTRI) grant # UL1TR001442. UCSD Moores Cancer Center Technology Shared Resource is supported by a National Cancer Center Support Grant (CCSG P30CA23100). We thank Dr. G. Stark for sharing U-series cell lines. T.D.T. was supported by NIH P30 DK063491 and NIH T32DK007044. M.L. was supported by T32CA009523. This study was supported by NIH R01 CA104509 and R01 CA 232147 to D-E.Z.

## Author contributions

K.-I.A., C.K.G., and D.-E.Z. designed and supervised the study. K.-I.A., S.M., and M.L. analyzed the scRNA-seq data. K.-I.A., S.M., T.D.T., and S.A.S. analyzed the RNA-seq data. T.D.T. analyzed the ATAC and H3K27ac ChIP-seq data. K.-I.A., S.M., and Y.Z. analyzed the mouse model data. K.-I.A., S.M., Y.Z., M.L., S.A.S., A.G.D., J.-B.F., Y.-J.H., and M.Y. contributed experimental data. K.-I.A., A.G.D., and D.-E.Z. wrote the manuscript, with input from all authors. All the authors critically reviewed the manuscript and approved the final version.

## Competing interests

The authors declare no competing interests.
