## [Peer Review File · Nature Communications]

Expansion of interferon inducible gene pool via USP18 inhibition promotes cancer cell pyroptosisEditorial Note: This manuscript has been previously reviewed at another journal that is not operating a transparent peer review scheme. This document only contains reviewer comments and rebuttal letters for versions considered at *Nature Communications*.

REVIEWER COMMENTS

Reviewer #1 (Remarks to the Author):

Here, Arimoto and colleagues report that heterozygosity of USP18 sensitises AML1-ETO9a-driven AML to interferon-induced cell death. The revised ms has much improved by toning down their language, including relevant literature and including a wealth of new data. Overall, I am satisfied with the revised ms and have only a few minor issues, as mentioned below.

Fig 1C: How many independent 'clones' of USP18+/fAE9a AML and USP18+/-AE9a AML were tested here. Is it only 1 transduction of BMDMs from 1 mouse, respectively? The authors should provide 3 independent biological replicates here.

Fig. 1e should include other immune cell populations, such as T-regs. At present it is not clear whether the increase in CD8+ T cells is selective, or whether this is accompanied with a general increase in immune cells (such as suppressor cells). A more detailed immune profiling will be required.

The result in Fig. 7i does not demonstrate that caspase-1-mediated GSDMD cleavage and IL-1b release occurs independent of cell death. It simply means that the kinetics of this event is superseded by another cell death event (caspase-8 mediated cleavage of GSDME). I suggest to tone down the language or, alternatively, conduct single cell analysis to evaluate whether the IL-1b releasing cell is alive or dead.

The authors used a plethora of KO cell lines. In most cases, the result of a single clone is shown. Can the authors demonstrate that multiple clones behave in the same way? Out of my own experience, there is a huge amount of clonal variation, hence it will be important to show the result of multiple clones.

Reviewer #2 (Remarks to the Author):

The authors have addressed most of my main comments in particular my concerns about the knockout mouse model. As such, I have no further major criticism.

Reviewer #4 (Replacement reviewer for Reviewer #3, Remarks to the Author):

Arimoto et al. demonstrate that deletion of USP18 in tumor cells strongly affects the cellular response to type IFN-I treatment. They show enhanced expression of ISGs leading to the induction of immunogenic cell death (pyroptosis). Enhanced expression of ISGs is expected, but the authors provide novel findings in terms of the transcriptional machinery that controls ISG expression and with regard to the repertoire of target ISGs. They show that USP18 is a negative regulator of STAT2 in the nucleus and that deletion of USP18 expands the set of STAT2 target genes, including "atypical" ISGs such as PLK2, which in turn is involved in pyroptosis induction. The majority of the data have been generated in a murine AML model and were confirmed in vitro on human cell lines THP-1 (AML) and MDA-MB-231 (breast cancer).

Comments of the reviewer:

Induction of immunogenic tumor cell death in a clinically highly relevant topic in cancer

immunotherapy, which is emphasized also by the authors in the abstract and the introduction of their manuscript.

The authors provide convincing mechanistic data regarding ICD (pyroptosis) of IFN-I-treated USP18-deficient tumor cells, but the therapy context is only barely addressed. There are a few data for the solid cancer models B16 (melanoma) and MC38 (colon) in supplemental Figure 9. However, important information is missing and analyses should be extended.

For the B16 model the following vaccines should be tested in parallel in tumor challenge experiments: B16-WT, IFN-I-treated B16-WT, B16-USP18^{-/-}, IFN-I-treated B16-USP18^{-/-}. So far, only studies with IFN-I-treated B16-USP18^{-/-} have been performed (in comparison to cisplatin-treated B16-USP18^{-/-} cells).

It should be determined which lymphocyte subset(s) mediate(s) the vaccine effect.

Similar vaccination experiments should be carried out in the MC38 model.

Such experiments are of critical importance as long as the authors place their novel findings in the context of cancer immunotherapy.

Minor:

In some figures the labeling (units) of the axis is missing. The labeling of other figures is hardly readable also not upon magnification (e.g. Fig. 7, Western blots, have lysates from USP18^{-/-} or USP18^{+/-} or USP18^{+/+} cells been tested? Information can be found in the legend but should also be readable in the figure).

Dear reviewers,

Thank you so much for your time and effort spent reviewing our manuscript. In the revised manuscript, we added data and modified text according to additional suggestions from you. We have addressed each of your comments point by point below:

Reviewer #1:

Here, Arimoto and colleagues report that heterozygosity of USP18 sensitises AML1-ETO9a-driven AML to interferon-induced cell death. The revised ms has much improved by toning down their language, including relevant literature and including a wealth of new data. Overall, I am satisfied with the revised ms and have only a few minor issues, as mentioned below.

We appreciate these positive comments about our revised manuscript.

Fig 1C: How many independent 'clones' of USP18+/fAE9a AML and USP18+/-AE9a AML were tested here. Is it only 1 transduction of BMDMs from 1 mouse, respectively? The authors should provide 3 independent biological replicates here.

The reviewer is right that data presented in Fig 1c were collected from one set of transduced cells in one mouse as a representative result. According to the suggestion, we added data from two additional independent sets (clones 2 and 3) of transduced cells in source data Fig 1C and modified the figure legend. Although there were clonal variations, all data showed *Usp18*^{+/-} AE9a AML has higher IFN sensitivity compared to *Usp18*^{+/-} BMDM as shown in the figure below.

Fig. 1e should include other immune cell populations, such as Tregs. At present it is not clear whether the increase in CD8+ T cells is selective, or whether this is accompanied with a general increase in immune cells (such as suppressor cells). A more detailed immune profiling will be required.

We appreciate the valuable comment. We redid the same experiment, and analyzed immune profiling in detail. As shown below, although the Tregs increase was not significant, the data showed that most types of immune cells were generally increased. Among them, the increase of activated T cells was indeed significant as we originally presented. We did not observe a significant increase of B cells. We updated the data for Fig 1e to include Tregs, B cells and more activated CD8⁺ data and modified the legend. Raw data for all other populations are also included in source data Fig 1e.

The result in Fig. 7i does not demonstrate that caspase-1-mediated GSDMD cleavage and IL-1b release occurs independent of cell death. It simply means that the kinetics of this event is superseded by another cell death event (caspase-8 mediated cleavage of GSDME). I suggest to tone down the language or, alternatively, conduct single cell analysis to evaluate whether the IL-1b releasing cell is alive or dead.

We agree and have toned down the statement on line 440-441 to 'although it remains possible that caspase-8-mediated cleavage of GSDME complements the cell death, which needs to be further investigated in future (Fig 7i)'.

The authors used a plethora of KO cell lines. In most cases, the result of a single clone is shown. Can the authors demonstrate that multiple clones behave in the same way? Out of my own experience, there is a huge amount of clonal variation, hence it will be important to show

the result of multiple clones.

We share the same concern with the reviewer about clonal variations. Therefore, we used at least two unique clones. We observed certain clonal variations but confirmed that the different KO clones have similar biological consequences for critical experiments (figures below). The associated raw data have been added to the source data file. In addition, we used shRNAs as a parallel approach to further validate the important findings obtained by knockouts, such as for USP18 and PLK2 (Extended data Fig 11 f-j).

Fig. 6d another clone data set

Fig. 7 b and c another clone data set

Fig. 7 n and o another clone data set

Fig. 8 b and c another clone data set

Extended data Fig. 9b another clone data set

Reviewer #2:

The authors have addressed most of my main comments in particular my concerns about the knockout mouse model. As such, I have no further major criticism.

We appreciate the positive feedback from this reviewer.

Reviewer #4 (Replacement reviewer for Reviewer #3):

Arimoto et al. demonstrate that deletion of USP18 in tumor cells strongly affects the cellular response to type IFN-I treatment. They show enhanced expression of ISGs leading to the induction of immunogenic cell death (pyroptosis). Enhanced expression of ISGs is expected, but the authors provide novel findings in terms of the transcriptional machinery that controls ISG expression and with regard to the repertoire of target ISGs. They show that USP18 is a negative

regulator of STAT2 in the nucleus and that deletion of USP18 expands the set of STAT2 target genes, including “atypical” ISGs such as PLK2, which in turn is involved in pyroptosis induction. The majority of the data have been generated in a murine AML model and were confirmed in vitro on human cell lines THP-1 (AML) and MDA-MB-231 (breast cancer).

Comments of the reviewer:

Induction of immunogenic tumor cell death in a clinically highly relevant topic in cancer immunotherapy, which is emphasized also by the authors in the abstract and the introduction of their manuscript.

The authors provide convincing mechanistic data regarding ICD (pyroptosis) of IFN-I-treated USP18-deficient tumor cells, but the therapy context is only barely addressed. There are a few data for the solid cancer models B16 (melanoma) and MC38 (colon) in supplemental Figure 9. However, important information is missing and analyses should be extended.

For the B16 model the following vaccines should be tested in parallel in tumor challenge experiments: B16-WT, IFN-I-treated B16-WT, B16-USP18^{-/-}, IFN-I-treated B16-USP18^{-/-}. So far, only studies with IFN-I-treated B16-USP18^{-/-} have been performed (in comparison to cisplatin-treated B16-USP18^{-/-} cells).

It should be determined which lymphocyte subset(s) mediate(s) the vaccine effect.

Similar vaccination experiments should be carried out in the MC38 model.

Such experiments are of critical importance as long as the authors place their novel findings in the context of cancer immunotherapy.

We appreciate the reviewer's valuable comments and highlighting of our novel findings.

In the original Extended Figure 9e, we aimed to emphasize that dying IFN-I-treated USP18^{+/-} cells have ICD ability in comparison to non-ICD dying cisplatin treated USP18^{+/-} cells by gold standard in vivo vaccination assay. We actually included B16-WT/IFN-I-treated B16-WT/B16-USP18^{+/-} in this setting. Unfortunately, these mice needed to be euthanized according to maximum tumor size limits. We also avoided removing the tumor at vaccination sites because it is still unclear whether surgical removal of primary tumor affects the tumor immune response. (10.3390/jcm9124096, 10.1158/0008-5472.can-03-2646, 10.4049/jimmunol.179.3.1960).

However, we agree with the reviewer's concern. As suggested by the reviewer, we were able to carry out such experiment using the MC38 model, including the MC38-WT and IFN-I-treated MC38-WT vaccine because of their relatively slower growth rate and high immunogenicity.

The results showed that tumor volume at the challenge site in the group of IFN-I-treated MC38-USP18^{+/-} vaccine was significantly reduced compared to the other vaccine groups, which is indicative of the higher ICD ability of IFN-I-treated MC38-USP18^{+/-} cells. (New extended data Fig.10c).

We also examined which subset(s) of lymphocyte mediate this vaccine effect. As below, we observed a significant increase of $CD8^+$ T cells and activated $CD8^+$ T cell infiltration in tumors in the IFN-I-treated MC38-USP18+/- vaccine group, which is a critical step in cancer-ICD vaccine mediated anti-tumor response. We also observed enhanced NK cell infiltration in the IFN-I-treated MC38-USP18+/- vaccine group. We observed enhanced Tregs but the $CD8^+$ /Tregs ratio on average is higher in the IFN-I-treated MC38-USP18+/- vaccine group. As seen in radiation induced ICD (doi: 10.1038/nrclinonc.2016.211), dying IFN-I-treated MC38-USP18+/- may also induce immunosuppressive effect as negative feedback. We did not see any change in B cell infiltration in tumor.

We have put these data in extended data fig 10d. Again, we highly appreciate the reviewer's comments, which have helped to improve our novel findings in the context of cancer immunotherapy.

Minor:

In some figures the labeling (units) of the axis is missing. The labeling of other figures is hardly readable also not upon magnification (e.g. Fig. 7, Western blots, have lysates from USP18^{-/-} or USP18^{+/-} or USP18^{+/+} cells been tested? Information can be found in the legend but should also be readable in the figure).

We thank the reviewer for pointing this out. We have checked throughout the manuscript and corrected any errors. We have also uploaded power point files with figures not embedded to word format.

REVIEWERS' COMMENTS

Reviewer #1 (Remarks to the Author):

The authors have made valuable changes and I am happy with their changes.

Reviewer #4 (Remarks to the Author):

Arimoto addressed most of my comments.

While the authors did not generate additional data in the B16 model (the given explanation is hard to understand), they carried out important experiments in the MC38 model demonstrating the relevance of their findings for cancer immunotherapy.

In the legend to suppl. Fig. 10f it is written: Tumor volume resulting from mice vaccinated with IFN β -treated USP18 $^{+/+}$ B16F10 or USP18 $^{+/-}$ MC38 cells subsequently challenged with live MC38 murine colon cancer cells.

Comment: B16F10 is not correct

Point-by-point response to the reviewer's comments

REVIEWERS' COMMENTS

Reviewer #1 (Remarks to the Author):

The authors have made valuable changes and I am happy with their changes.

We appreciate the positive comment from this reviewer.

Reviewer #4 (Remarks to the Author):

Arimoto addressed most of my comments.

While the authors did not generate additional data in the B16 model (the given explanation is hard to understand), they carried out important experiments in the MC38 model demonstrating the relevance of their findings for cancer immunotherapy.

In the legend to suppl. Fig. 10f it is written: Tumor volume resulting from mice vaccinated with IFN β -treated USP18 $^{+/+}$ B16F10 or USP18 $^{+/-}$ MC38 cells subsequently challenged with live MC38 murine colon cancer cells.

Comment: B16F10 is not correct

We thank the positive feedback and the comment from this reviewer. We corrected the supplementary figure 10f's legend.